# Terminal uranium(V)-nitride hydrogenations involving direct addition or Frustrated Lewis Pair mechanisms

Lucile Chatelain [1], Elisa Louyriac[2], Iskander Douair[2], Erli Lu [1], Floriana Tuna [3], Ashley J. Wooles[1], Benedict M. Gardner[1], Laurent Maron [2]* & Stephen T. Liddle [1]*

Despite their importance as mechanistic models for heterogeneous Haber Bosch ammonia synthesis from dinitrogen and dihydrogen, homogeneous molecular terminal metal-nitrides are notoriously unreactive towards dihydrogen, and only a few electron-rich, low-coordinate variants demonstrate any hydrogenolysis chemistry. Here, we report hydrogenolysis of a terminal uranium(V)-nitride under mild conditions even though it is electron-poor and not low-coordinate. Two divergent hydrogenolysis mechanisms are found; direct 1,2-dihydrogen addition across the uranium(V)-nitride then H-atom 1,1-migratory insertion to give a uranium (III)-amide, or with trimesitylborane a Frustrated Lewis Pair (FLP) route that produces a uranium(IV)-amide with sacrificial trimesitylborane radical anion. An isostructural uranium (VI)-nitride is inert to hydrogenolysis, suggesting the $5f^1$ electron of the uranium(V)-nitride is not purely non-bonding. Further FLP reactivity between the uranium(IV)-amide, dihydrogen, and triphenylborane is suggested by the formation of ammonia-triphenylborane. A reactivity cycle for ammonia synthesis is demonstrated, and this work establishes a unique marriage of actinide and FLP chemistries.

[1] Department of Chemistry, The University of Manchester, Oxford Road, Manchester M13 9PL, UK. [2] LPCNO, CNRS & INSA, Université Paul Sabatier, 135 Avenue de Rangueil, Toulouse 31077, France. [3] Department of Chemistry and Photon Science Institute, The University of Manchester, Oxford Road, Manchester M13 9PL, UK. *email: laurent.maron@irsamc.ups-tlse.fr; steve.liddle@manchester.ac.uk

Terminal metal-nitrides, M≡N, represent a key fundamental class of metal-ligand linkage in coordination chemistry[1]. Although these M≡N triple bonds have been of elementary interest for over 170 years[2], only in relatively recent times has there been a concerted effort to study their reactivity[1,3]. However, although a variety of reactivity patterns have emerged with metal-nitrides[1], the vast majority are remarkably unreactive because strong, often highly covalent M≡N triple bonds that result from high oxidation state metal ions—needed to bind to the hard, charge-rich nitride, $N^{3-}$—renders them inherently inert[1,3]. One strategy to increase the reactivity of metal-nitrides is to utilise low oxidation state electron-rich metals to destabilise the M≡N triple bond, but by definition such metals are ill-matched to nitrides and so are difficult to prepare[4]. Additionally, reactivity of metal-nitrides often involves ancillary ligands rather than the M≡N triple bond itself. Overcoming this challenge is difficult because there are very few metal-nitrides where the metal oxidation state or co-ligands can be varied within a homologous family to encourage M≡N triple bond reactivity[1,5].

Since there is an isoelectronic relationship between the M≡N and N≡N triple bonds of metal-nitrides and dinitrogen, $N_2$, respectively, the former are fundamentally mechanistically important with respect to Haber Bosch chemistry where they are invoked as intermediates in the cleavage of the latter and conversion to ammonia, $NH_3$, by hydrogenolysis with dihydrogen, $H_2$[6,7]. There has thus been intense interest in the reactivity of metal-nitrides with $H_2$, and indeed their use in N-atom transfer reactivity and catalysis more widely[8–12], but there are few reports of molecular metal-nitrides reacting with $H_2$, and indeed activating $H_2$ in this homogeneous context remains a significant challenge in contrast to heterogeneous Haber Bosch chemistry where $H_2$-cleavage is essentially barrier-less[6]. One solution to overcome this hydrogenolysis challenge may be to exploit Frustrated Lewis Pair (FLP) chemistry[13,14], but so far this has been focussed on M–$N_2$ complexes[15,16]. Usually with mid- or late-transition metals[3], most metal-nitride hydrogenations involve sequential protonations[17–22], but bridging nitrides in poly-iron-/titanium-/zirconium complexes have been reported to react with $H_2$ to give imido-hydride and $NH_3$ products[23–25]. Only three terminal metal-nitrides have been reported to undergo hydrogenolysis with $H_2$. The isostructural $d^4$ ruthenium(IV)- and osmium(IV)-nitrides $[M\{N(CH_2CH_2PBu^t_2)_2\}(N)]$ (M = Ru, Os) react with $H_2$ using the ancillary ligand to shuttle H-atoms to evolve $NH_3$[26,27], and the $5d^6$ iridium(III)-nitride $[Ir\{NC_5H_3\text{-}2,2'\text{-}(C[Me]=N\text{-}2,6\text{-}Pr^i_2C_6H_3)_2\}(N)]$ undergoes concerted reactivity with $H_2$ to give $[Ir\{NC_5H_3\text{-}2,2'\text{-}(C[Me]=N\text{-}2,6\text{-}Pr^i_2C_6H_3)_2\}(NH_2)]$[28]. Thus, direct hydrogenolysis of a M≡N triple bond with $H_2$ remains exceedingly rare, and involves reasonably electron-rich ($\geq d^4$) metal complexes with low coordination numbers.

As part of our studies investigating actinide-ligand multiple bonding supported by triamidoamine ancillary ligands[29–35], we have reported two closely related terminal uranium-nitrides $[U^V(Tren^{TIPS})(N)][K(B15C5)_2]$ (1) and $[U^{VI}(Tren^{TIPS})(N)]$ (2) $[Tren^{TIPS} = N(CH_2CH_2NSiPr^i_3)_3{}^{3-}$; B15C5 = benzo-15-crown-5 ether][36–38] that, unusually[1,5,39], permit examination of the electronic structure and reactivity of the same isostructural terminal nitride linkage with more than one metal oxidation state. Both react with the small molecules CO, $CO_2$, and $CS_2$[40,41], but since only the protonolysis of 1 with $H_2O$ to give $NH_3$ had been previously examined[36] the ability of 1 and 2 to react with $H_2$ has remained an open question. Indeed, the study of molecular uranium-nitride reactivity remains in its infancy[36,37,40–48], and only very recently the diuranium(IV)-nitride-cesium complex $[Cs\{U(OSi[OBu^t]_3)_3\}_2(\mu\text{-}N]$ was reported to reversibly react with $H_2$ to give the diuranium-imido-hydride complex $[Cs\{U(OSi[OBu^t]_3)_3\}_2(\mu\text{-}NH)(\mu\text{-}H)]$[47]. Bridging nitrides tend to be more

reactive than terminal ones, so whilst this nitride hydrogenolysis is enabled by the bridging nature of the nitride and polymetallic cooperativity effects[47], we wondered whether $H_2$ activation by 1 or 2 might still be accessible, given prior protonation studies[36], since this would realise the first terminal f-block-nitride hydrogenolyses. Further motivation to study this fundamental reaction stems from the fact that bridging and terminal uranium-nitride reactivity with $H_2$ is implicated in Haber Bosch $NH_3$ synthesis when uranium is used as the catalyst[49], and uranium-nitrides have been proposed as accident tolerant fuels (ATFs) for nuclear fission, but likely reactivity with $H_2$ formed from radiolysis under extreme conditions or when stored as spent fuel remains poorly understood.

Here, we report that 2 does not react with $H_2$ consistent with a strong U≡N triple bond that is inherently unreactive like many high oxidation state terminal metal-nitrides. However, in contrast 1 reacts with $H_2$ under mild conditions despite the fact it can be considered to be a high oxidation state metal and not of a low coordination number nor electron-rich as a $5f^1$ metal ion. This hydrogenolysis reactivity is thus unprecedented in molecular metal-nitride chemistry, and further supports the emerging picture that suggests that the 5f-electron of 1 should not be considered as purely nonbonding. This study reveals two distinct $H_2$-activation mechanisms. When the borane $BMes_3$ (Mes = 2,4,6-trimethylphenyl) is present a FLP mechanism operates where two $H_2$ heterolysis events and a borane reduction step sequentially combine to furnish a $U^{IV}\text{-}NH_2$ product, and this, to the best of our knowledge, is the first demonstration of the application of bona fide FLP reactivity to actinide chemistry. When the borane is absent, direct 1,2-addition of $H_2$ across the U≡N triple bond to give a H−$U^V$=N−H intermediate followed by H-atom migration produces a $U^{III}\text{-}NH_2$ product that is easily oxidised to $U^{IV}\text{-}NH_2$. The direct addition is slower than the FLP-mediated mechanism, demonstrating the facilitating role of FLPs. We find evidence that treating the $U^{IV}\text{-}NH_2$ product with $BPh_3$ and $H_2$ produces further FLP hydrogenolysis reactivity, since $H_3NBPh_3$ has been detected in reaction mixtures, but this is reversible and produces products that react to give the starting materials. While currently of no practical use this demonstrates further potential for FLPs in this area. We demonstrate an azide to nitride to amide to ammonia reaction cycle, supported by overall hydrogenation involving hydrogenolysis and electrophilic quenching steps.

## Results

**Hydrogenolysis of the terminal uranium(V)-nitride bond.** Since 2 was found to be unreactive or decomposed to a complex mixture of intractable products when exposed to boranes in the context of this study we examined the reactivity of 1. With or without $H_2$, treatment of 1 in toluene with the strong Lewis acid $B(C_6F_5)_3$ (BCF) results in decomposition as evidenced by $^{19}F$ NMR spectra of reaction mixtures that show multiple fluorine resonances consonant with multiple C–F activation reactions, Fig. 1. Deleterious C–F bond activation reactivity is well documented for BCF[50], and so we examined the reaction of 1 with the less Lewis acidic $BPh_3$. However, when 1 is treated with $BPh_3$ in toluene the adduct complex $[U^V(Tren^{TIPS})(NBPh_3)][K(B15C5)_2]$ (3), which when compared to 1 and 2 is perhaps best formulated as a uranium(V)-imido-borate rather than a uranium(V)-nitrido-borane, is rapidly formed quantitatively and isolated in crystalline form in 66% yield, Fig. 1.

The retention of uranium(V) in 3 is supported by absorptions in the 5000–12,500 $cm^{-1}$ region of its UV/Vis/NIR spectrum (Supplementary Fig. 1) that are characteristic of intraconfigurational $^2F_{5/2}$ to $^2F_{7/2}$ transitions of uranium(V)[38,51], and by variable-temperature SQUID magnetometry, Fig. 2 and Supplementary

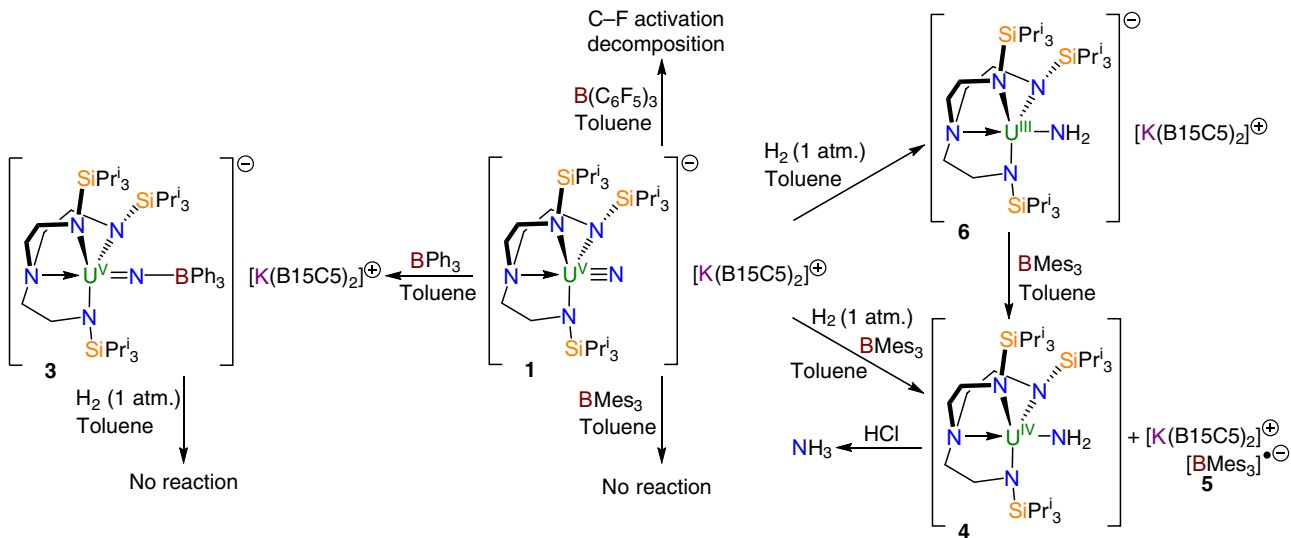

**Fig. 1 Synthesis of complexes 3–6.** Treatment of **1** with the strong Lewis acid $B(C_6F_5)_3$ results in decomposition, however the milder borane $BPh_3$ produces the capped species **3**, which is inert with respect to reaction with $H_2$. Complex **1** does not react with the sterically encumbered $BMes_3$, but exposure of that mixture to $H_2$ produces the amide complex **4** with concomitant formation of the radical anion complex **5**. Addition of $H_2$ to **1** produces the amide complex **6**, and subsequent treatment with $BMes_3$ produces **4** and **5**. Treating **4** with HCl produces $NH_3$. B15C5 = benzo-15-crown-5 ether. Mes = 2,4,6-trimethylphenyl.

Fig. 2. A powdered sample of **3** returns a magnetic moment of 2.23 $\mu_B$ at 300 K (1.96 $\mu_B$ by solution Evans method) that changes little until 30 K where it falls quickly to a moment of 1.38 $\mu_B$ at 2 K and this is consistent with the magnetic doublet character of $5f^1$ uranium(V)[52–54]. The solid-state structure of **3**, Fig. 3 and Supplementary Fig. 3, reveals a separated ion pair formulation where the nitride has been capped by the $BPh_3$ unit. The U–N$_{imido}$ bond length of 1.911(6) Å is consistent with the imido-borate formulation, for example distances of 1.916(4), 1.954(3), and 1.946(13) Å are found in $[(Bu^tArN)_3U^V(NBCF)][NBu^n_4]$ (Ar = 3,5-dimethylphenyl)[55], $[U^V(Tren^{TIPS})(NSiMe_3)]$[37], and $[U^V(Tren^{TIPS})(NAd)]$ (Ad = 1-adamantyl)[37], respectively, and the B-N$_{imido}$ distance of 1.581(9) Å compares well to the sum of the single bond covalent radii of B and N (1.56 Å)[56]. The U–N$_{amine}$ distance of 2.737(5) Å is long, reflecting the dative nature of the amine donor and that it is *trans* to the strong imido donor, and the U–N$_{amide}$ distances (2.254(7)-2.312(6) Å) are slightly long for such distances[57], reflecting the formal anionic nature of the uranium component of **3**.

Complex **3** does not react with $H_2$ (1 atm.), Fig. 1. Indeed, dissolving a mixture of **1** and $BPh_3$ under $H_2$ only generates **3**, and so since $BPh_3$ has shut all reactivity down by strongly binding to the nitride of **1**, but BCF is too reactive, we examined the use of $BMes_3$ (Mes = 2,4,6-trimethylphenyl). In principle, the *ortho*-methyls of the Mes groups of this borane block deactivating strong coordination of Lewis bases to the vacant p-orbital of boron whilst retaining a Lewis acidic boron centre.

To provide a reactivity control experiment, we stirred a 1:1 mixture of **1**:$BMes_3$ in toluene under an atmosphere of $N_2$ and find no evidence for any adduct formation, Fig. 1, with only free $BMes_3$ being observed as evidenced by a resonance at 76.8 ppm in the $^{11}B$ NMR spectrum of the reaction mixture. Repeating this reaction, but under $H_2$ (1 atm.), over two days at 298 K results in complete consumption of starting materials with deposition of a dark blue solid. The brown supernatant was removed and found by NMR spectroscopy to contain the known uranium(IV)-amide $[U^{IV}(Tren^{TIPS})(NH_2)]$ (**4**) in 67% yield, Fig. 1, as evidenced by a resonance at 107 ppm in its $^1H$ NMR spectrum that corresponds to the amide protons[37]. A control experiment, stirring **1** in toluene over two weeks, also produces **4** from trace, adventitious

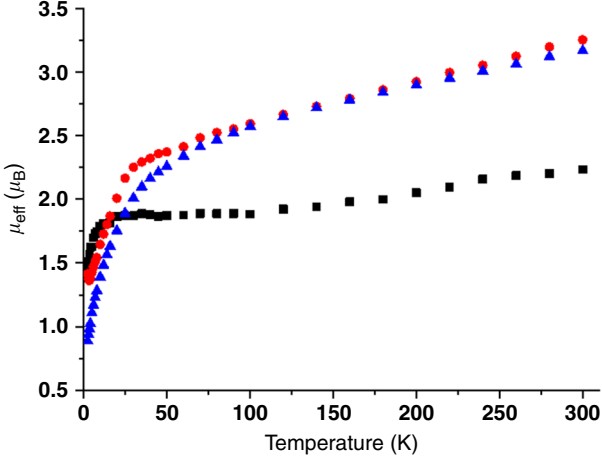

**Fig. 2 Variable-temperature SQUID magnetic moment data for 3, 6, and 8.** Key: **3** (black squares), **6** (red circles), and **8** (blue triangles). Data were measured in an applied magnetic field of 0.1 Tesla.

sources of $H^+$, though in far lower proportions, so to prove that the source of H-atoms in **4** originates from $H_2$, and not adventitious $H^+$[58], the reaction was repeated under $D_2$ (1 atm., 99.8% atom D). Interestingly, whilst $[U^{IV}(Tren^{TIPS})(ND_2)]$ (**4″**, $^2H$ δ 107.5 ppm) is formed, confirming that hydrogenolysis by $H_2/D_2$ does indeed occur, it is always accompanied by **4** and $[U^{IV}(Tren^{TIPS})(NHD)]$ (**4′**, $^1H$ δ 106 ppm, $^2H$ δ 106.8 ppm, $^2J_{HD}$ not resolved). This reveals that H/D exchange occurs over time, so to determine the source of this exchange we studied the reaction of **1** and $BMes_3$ with all combinations of $H_2/D_2$ with $H_6$-/$D_6$-benzene and $H_8$-/$D_8$-toluene (see Supplementary Methods). We find that when $H_2$ is used only **4** is ever detected, but when $D_2$ is used **4**, **4′**, and **4″** all form (av. 12, 24, and 64%, respectively) irrespective of whether the solvent is deuterated or not which rules out arene solvents as the H-source. However, when using $C_6H_6$ as solvent for the reaction of **1** with $BMes_3$ and $D_2$ a weak resonance is observed at −5.2 ppm in the $^2H$ NMR spectrum (*cf* −5.35 and −5.87 ppm for *iso*-propyl methine and

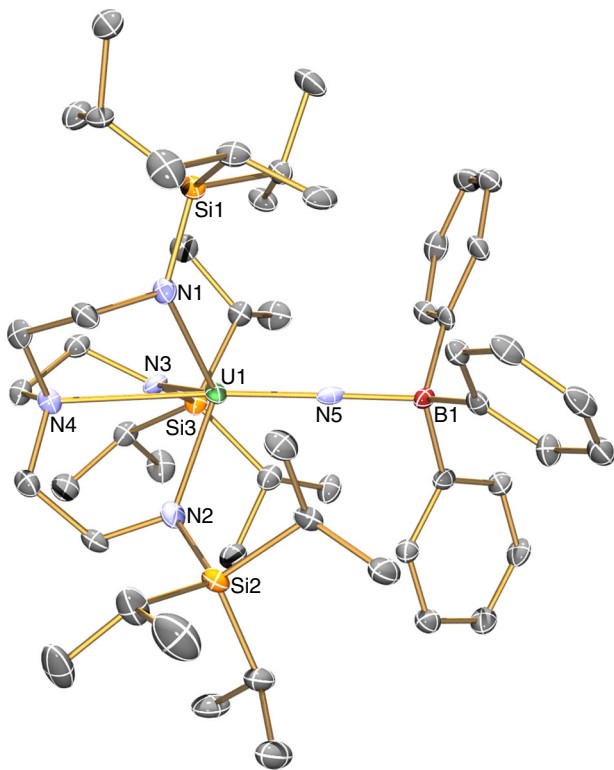

**Fig. 3 Molecular structure of the anion component of 3 at 150 K and displacement ellipsoids set to 40%.** Hydrogen atoms, minor disorder components, lattice solvent, and the $[K(B15C5)_2]^+$ cation component are omitted for clarity. Selected bond lengths (Å): U1-N1, 2.305(5); U1-N2, 2.254 (7); U1-N3, 2.312(6); U1-N4, 2.737(5); U1-N5, 1.911(6); B1-N5, 1.581(9).

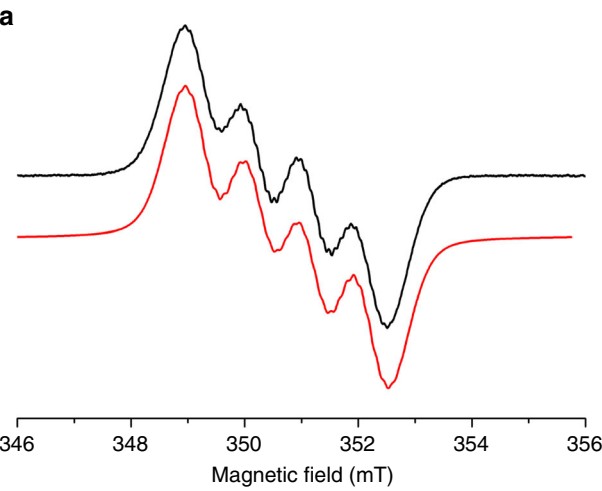

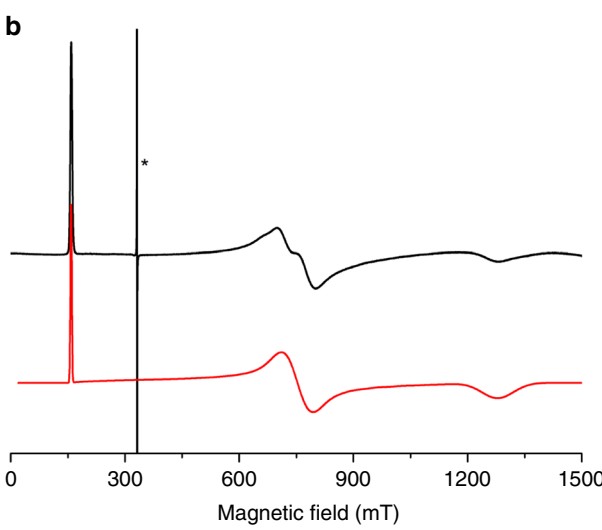

**Fig. 4 EPR data for 5 and 6. a** X-band (9.8 GHz) EPR spectrum of **5** at 298 K as a 9 mM solution in THF. The black line is the experimental spectrum and the red line the simulation with $g = 2.003$, $A(^{11}B) = 9.44$ G, $A(^{10}B) = 2.7$ G, $A(^1H) = 1.2$–1.4 G. **b** X-band (9.3 GHz) EPR spectrum of a powdered sample of **6** at 20 K. The black line is the experimental spectrum and the red line the simulation with $g = 4.19$, 0.88, and 0.52. The very sharp signal marked with asterisk is a very small quantity of radical impurity with $g = 2$.

methyl protons, respectively, in the $^1H$ NMR spectrum of **4**). We therefore suggest that the H-source is the Tren$^{TIPS}$ Pr$^i$ groups since they have precedent for forming cyclometallates[57], a reversible amide/imido-cyclometallate + H$_2$ equilibrium can be envisaged since it has been previously shown that uranium-Tren-cyclometallates can react reversibly with H$_2$/D$_2$[59], and this would also account for the absence of D-scrambling into **4** since Tren$^{TIPS}$ is void of D-atoms.

The dark blue solid was isolated and after work-up obtained as dark blue crystals, identified as the radical species [K(B15C5)$_2$][BMes$_3$] (**5**), in 69% yield. This compound has been structurally characterised by single crystal diffraction, see Supplementary Fig. 4. Compound **5** is very similar to [Li(12-crown-4)$_2$][BMes$_3$][60] that contains the same radical anion component, and the EPR data of **5** ($g = 2.003$, $A(^{11}B) = 9.44$ G, $A(^{10}B) = 2.7$ G, $A(^1H) = 1.2$–1.4 G), Fig. 4a, confirm the formation of the BMes$_3^{•-}$ radical anion formulation. The UV/Vis/NIR spectrum of **5** exhibits an intense, ($\varepsilon = \sim 8000$ M$^{-1}$ cm$^{-1}$) broad absorption centred at $\sim 12,800$ cm$^{-1}$, which largely accounts for its dark blue colour, see Supplementary Fig. 5.

Since **1** does not form an adduct with BMes$_3$, but the introduction of H$_2$ leads to hydrogenolysis to give the amide **4**, we surmised that the **1**/BMes$_3$ mixture may constitute a Frustrated Lewis Pair (FLP) system that is evidently capable of activating H$_2$, which is confirmed computationally (see below). However, since H$_2$ can be a two-electron reducing agent and uranium(V) is normally quite oxidising, we hypothesised that the BMes$_3$ may not actually be required. In effect, when **4** is produced it is essentially at the expense of the sacrificial one-electron reduction of BMes$_3$ to BMes$_3^{•-}$, which would formally invoke a uranium(III)-amide precursor that would be nicely in-line with a H$_2$-uranium(V)

two-electron redox couple. In order to test whether the FLP aspect of this hydrogenolysis chemistry is vital to effecting dihydrogen activation a toluene solution of **1** under H$_2$ (1 atm.) was stirred without BMes$_3$. Over seven days **1** is consumed with concomitant precipitation of a gray solid identified as the uranium(III)-amide [U$^{III}$(Tren$^{TIPS}$)(NH$_2$)][K(B15C5)$_2$] (**6**) (45% yield), Fig. 1. The hydrogenolysis reaction is now slower than when BMes$_3$ is present, but the reaction is best conducted at 288 and not 298 K, which may also retard the rate of reactivity. When the reaction is alternatively conducted under D$_2$ (1 atm.), a mix of **6**, [U$^{III}$(Tren$^{TIPS}$)(NHD)][K(B15C5)$_2$] (**6′**) and [U$^{III}$(Tren$^{TIPS}$)(ND$_2$)][K(B15C5)$_2$] (**6″**) are isolated (66% yield by uranium content) analogously to **4/4′/4″**, again indicating H/D exchange but confirming the H-atoms of the amide unit in **6** originate from gaseous H$_2$. Consistent with these observations, we find that **1** also reacts with 9,10-dihydroanthracene (pKa 31 in DMSO, *cf* 34±4 for H$_2$ in DMSO)[61] to produce an insoluble precipitate and **4** in solution. From this solution we isolated a small crop of red crystals formulated by $^1H$ NMR spectroscopy and X-ray

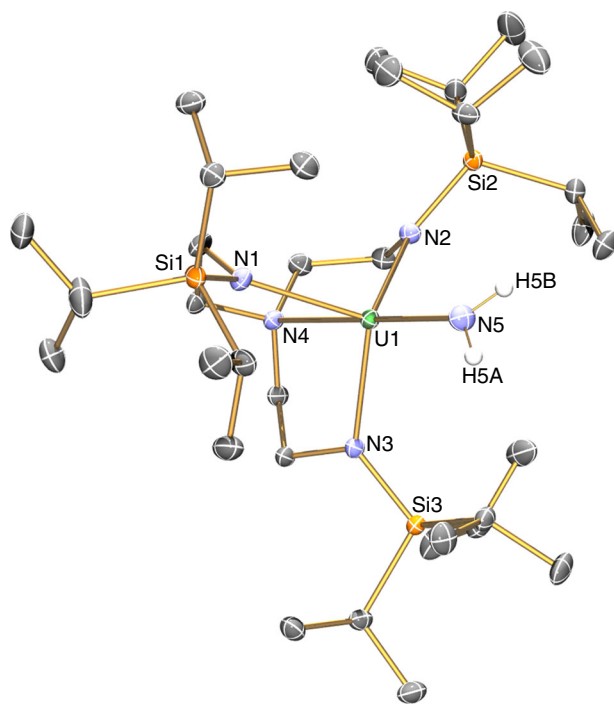

**Fig. 5 Molecular structure of the anion component of 6 at 150 K and displacement ellipsoids set to 40%.** Nonamide hydrogen atoms and the $[K(B15C5)_2]^+$ cation component are omitted for clarity. Selected bond lengths (Å): U1-N1, 2.393(2); U1-N2, 2.404(2); U1-N3, 2.385(2); U1-N4, 2.721(2); U1-N5, 2.335(3).

diffraction (see Supplementary Fig. 6) as $[K(B15C5)_2][C_{14}H_{11}]$. We suggest that **1** is converted to **6**, and this oxidises to **4** with concomitant reduction of anthracene, ultimately producing $[K(B15C5)_2][C_{14}H_{11}]$ via proton abstraction from solvent.

Unfortunately, **6/6′/6″** are highly insoluble in non-polar solvents and decompose in polar media so NMR and UV/Vis/NIR data could not be obtained. Complexes **6/6′/6″**, as their trivalent formulations suggest, are easily oxidised, and the mother liquor from these reactions always contains variable quantities of **4/4′/4″**, respectively, and heating suspensions of **6/6′/6″** in $C_6D_6$ in an attempt to obtain $^1H$ NMR spectra results in extraction of **4/4′/4″**, respectively. However, the $5f^3$ uranium(III) formulation of **6** is confirmed by variable-temperature SQUID magnetometry, Fig. 2 and Supplementary Fig. 7, where the magnetic moment of **6** is 3.25 $\mu_B$ at 300 K and this slowly decreases to 2.0 $\mu_B$ at ~20 K and then falls to 1.59 $\mu_B$ at 2 K[52–54]. Furthermore, the X-band EPR spectrum of **6** at 20 K, Fig. 4b, exhibits $g$ values of 4.19, 0.88, and 0.52, from which a magnetic moment of 2.16 $\mu_B$ would be predicted that is in good agreement with the observed magnetic moment of **6** at 20 K. The solid-state structure of **6** has been determined, Fig. 5 and Supplementary Fig. 8, revealing a separated ion pair formulation. The salient feature of **6** is the presence of a $U^{III}-NH_2$ linkage within the uranium component, which has no precedent in uranium(III) chemistry, as evidenced by a U–$N_{amide}$ distance of 2.335(3) Å, which is longer than analogous $U^{IV}-NH_2$ distances of 2.228(4) Å in **4**[37], 2.217(4) Å in $[U^{IV}\{\eta^8-C_8H_6-1,4-(SiPr^i_3)_2\}(\eta^5-C_5Me_5)(NH_2)]$[62], and 2.183(6) and 2.204(6) Å in $[U^{IV}\{\eta^5-C_5H_2-1,2,4-(Bu^t)_3\}(NH_2)_2]$[63]. The Tren U–$N_{amine}$ and U–$N_{amide}$ distances of 2.721(2) and 2.385(2)-2.393(2) Å, respectively, reflect the anionic uranium(III) formulation of **6**, since, for example, the latter, which are usually quite sensitive to the oxidation state of uranium, are usually ~2.27 Å for uranium(IV) congeners[57].

In order to experimentally link **6–4** we treated **6** with one equivalent of $BMes_3$ resulting in immediate reduction of $BMes_3$ to give a 1:1 mixture of **4** and **5**, Fig. 1, which is in-line with the reducing nature of **6** as evidenced by ready formation of **4** in supernatant reaction mixtures. These reactions show that although the FLP aspect of the reaction of $H_2$ with **1** certainly facilitates and accelerates the hydrogenolysis of the nitride linkage, it is not essential, and the terminal uranium(V)-nitride linkage is reactive enough in its own right to be hydrogenated with $H_2$ to give a uranium(III)-amide, and this is confirmed computationally (see below).

**Ammonia synthesis via strong acid.** After the hydrogenolysis reactions that produce **4** and **6** we vacuum transferred volatile materials onto hydrochloric acid, but in each case no more than a 5% yield of $NH_3$, as its conjugate acid $NH_4^+$, was detected by standard methods. This suggests that although the $U^V\equiv N$-nitride linkage reacts with one equivalent of $H_2$ to give $U^{III/IV}-NH_2$, further reaction of the latter linkages with $H_2$ does not occur. Direct treatment of **4′/4″** with 0.01 M HCl in THF/$Et_2O$, to differentiate the $D^+$ as from $D_2$ and not $D^+$ acid, vacuum transfer onto a 2 M HCl in $Et_2O$ acid trap, then assay, revealed a mixture of $NH_3D^+$ ($^2D$ δ 7.12 ppm) and $NH_4^+$ ($^1H$ δ 7.28 ppm, 1:1:1 triplet, $J_{NH} = 51$ Hz) by $^1H$ NMR spectroscopy. Addition of $H_2O$ results in full D/H exchange to give $NH_4^+$ as the sole ammonium species in 52% yield. Analogously, **6′/6″** produces $NH_4^+$ in 46% yield, and if the HCl acid steps are replaced with analogous DCl reagents then $NHD_3^+$ is first obtained and when this is converted to $NH_4^+$ a similar yield of 48% is obtained showing the internal consistency of this approach, Supplementary Figs. 9, 10. Under the action of strong acid the main by-product is $Tren^{TIPS}H_3$ from over-protonation, but up to 31% $[U^{IV}(Tren^{TIPS})(Cl)]$ (**7**)[36] could be observed by $^1H$ NMR spectroscopy as would be expected from the reaction of **4** with HCl.

**Reversible ammonia-borane formation.** Since $H_2$ does not react with **4** or **6** on their own, we examined whether addition of a borane would facilitate a second hydrogenolysis step; utilising BCF or $BMes_3$ with $H_2$ results in no reaction and/or formation of unknown, intractable products. We find, however, that **4** reacts with $BPh_3$ to form the uranium(IV)-amide-borane adduct $[U^{IV}(Tren^{TIPS})(NH_2BPh_3)]$ (**8**), Fig. 6, as evidence by its solid-state structure, Fig. 7. The salient feature of the structure of **8** is that although the Tren U–$N_{amine}$ and U–$N_{amide}$ distances of 2.645(5) and 2.221(5)-2.246(5) Å are unexceptional for Tren-uranium(IV) distances[57], the U–$NH_2$ U–$N_{amide}$ distance of 2.578(5) Å is very long[64], suggesting that coordination of $BPh_3$ has severely weakened the U–$NH_2$ linkage. However, there is clearly a balance of steric clashing in this region of the molecule since the B-$N_{amide}$ distance of 1.637(9) Å is ~0.06 Å longer than the analogous distance in **3** and ~0.08 Å longer than the sum of the covalent single bond radii of B and N (1.56 Å)[56]. Variable-temperature SQUID magnetometry on a powdered sample of **8**, Fig. 2 and Supplementary Fig. 11, confirms the uranium(IV) formulation of this complex. Specifically, the magnetic moment of **8** at 300 K is 3.17 $\mu_B$ and this decreases smoothly to a value of 0.89 $\mu_B$ at 2 K and is tending to zero[52–54]. This is characteristic of uranium(IV) which is a magnetic singlet at low temperature but that exhibits a small contribution from temperature independent paramagnetism to give a nonzero magnetic moment.

The $^{11}B$ NMR spectrum of **8** dissolved in $C_6D_6$ or $C_6D_5CD_3$ at 293 K exhibits resonances at 67.5 and −3.2 ppm, corresponding to free $BPh_3$ and $H_3NBPh_3$, respectively, as confirmed by comparison to authentic samples. The implication, consistent with the long U–$N_{amide}$ and B–$N_{amide}$ distances in **8**, is that **8** is in

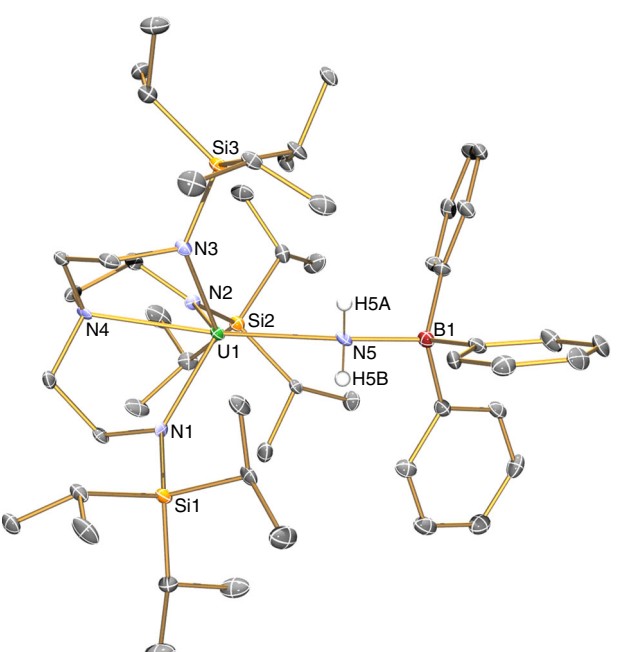

**Fig. 6 Reactivity of 4.** Treatment of **4** with BPh₃ results in the formation of the adduct **8**. Addition of H₂ to **8** produces the hydride **10** with concomitant elimination of H₃NBPh₃. Complex **10** eliminates H₂ to produce the cyclometallate complex **9**, which in turn can react with H₃NBPh₃ to reform **4**.

**Fig. 7 Molecular structure of 8 at 150 K and displacement ellipsoids set to 40%.** Nonamide hydrogen atoms and lattice solvent are omitted for clarity. Selected bond lengths (Å): U1-N1, 2.236(5); U1-N2, 2.221(5); U1-N3, 2.246(4); U1-N4, 2.645(5); U1-N5, 2.578(5); B1-N5, 1.637(9).

equilibrium with **4** and free BPh₃ in solution by B–N bond cleavage, but also that the long U–N$_{amide}$ bond is weakened increasing the basicity of this amide resulting in its rupture, C–H bond activation, and N–H bond formation to concomitantly form H₃NBPh₃ and the uranium(IV)-cyclometallate complex [U$^{IV}${N(CH₂CH₂NSiPr$^i$₃)₂(CH₂CH₂NSiPr$^i$₂CH(Me)CH₂)}] (**9**)[65]. Indeed, trace resonances that match reported data[65] for **9** could be observed. A variable-temperature ¹H and ¹¹B{¹H} NMR study (Supplementary Figs. 12, 13) reveals that at 293 K the dominant products are **4** and free BPh₃, but as the temperature is lowered to

253 and then 233 K resonances attributable to **8** grow in as **4** diminishes such that at 253 K the ratio of **4:8** is ~2:1 and at 233 K this ratio is ~2:3. However, when **9** is treated with H₃NBPh₃ the formation of **4** and BPh₃ are observed by ¹H NMR spectroscopy. This suggests facile, unspecific reversible reactivity but also hints at FLP-type reactivity, so we dissolved a 1:1 mixture of **4** and BPh₃ under H₂ (1 atm.), but again find only trace quantities of H₃NBPh₃. If **8** reacts with H₂ to form H₃NBPh₃ and [U$^{IV}$(Tren$^{TIPS}$)(H)] (**10**) the latter would be anticipated to eliminate H₂ to give cyclometallate **9**[59,65]. Indeed, treating [U$^{IV}$(Tren$^{TIPS}$)(THF)][BPh₄] with NaHBPh₃ to nominally produce [U$^{IV}$(Tren$^{TIPS}$)(HBPh₃)] gives H₂, BPh₃, and **9** in addition to the anticipated NaBPh₄ by-product. The reaction cycle in Fig. 6 can thus be proposed where **4** reacts with BPh₃ and H₂ to give, possibly via **8**, H₃NBPh₃ and **10**, the latter of which extrudes H₂ to give **9**. Since it is known that **9** reacts with H₃NBPh₃ to give **8** and/or **4** and free BPh₃ then a cycle is most likely established where reactivity is occurring but no discernable products can be isolated since the products react with one another to give the starting materials. Though of little use currently, the formation of H₃NBPh₃ suggests that it may be possible to extract out and trap the NH₃, though so far this system has resisted attempts to do so.

**Closing an ammonia synthesis reaction cycle.** Having effected hydrogenolysis of **1** but found that further reaction with H₂ either does not occur or seems to occur in a borane-cycle with no discernable products, we sought to close a reaction cycle utilising an electrophile. Accordingly, treatment of **4**, either prepared directly from **1**/H₂/BMes₃ or stepwise via **6**, with Me₃SiCl produces **7** and Me₃SiNH₂ that can be quantitatively converted to NH₃ in the form of ammonium salts. Under nonoptimised conditions an equivalent NH₃ yield of 53% was achieved. Thus, a reaction cycle for azide to nitride to amide to ammonia by hydrogenation overall is demonstrated at uranium using hydrogenolysis of H₂ followed by an electrophilic elimination and acid quench, Fig. 8.

**Computational reaction mechanism profiles.** In order to understand the reactions that produce **4/5** and **6**, DFT calculations (B3PW91) corrected for dispersion- and solvent-effects were carried out to determine possible reaction pathways for the reaction of complex **1** with H₂ in the presence or absence of BMes₃, Supplementary Tables 1–25. We also computed the reaction profile for the hypothetical reaction of **2** with H₂ (Supplementary Fig. 14), which confirms the experimental situation of no observable reactivity of **2** with H₂. In the absence of BMes₃, Fig. 9, H₂ reacts with **1** in a σ-bond metathesis fashion. The associated barrier is relatively low (14.4 kcal mol⁻¹). At the transition state (**B**), the H–H bond is strongly elongated (1.02 Å) and the N–H bond is not yet formed (1.35 Å). The U–N$_{nitride}$ bond is 1.84 Å and the U–H distance is long (2.20 Å). The N–H–H angle is 146.3°, which is quite acute for a metathesis reaction. The NPA charges at the transition state (TS) [U, +1.12; N, −0.84; H, +0.23; H, −0.10] indicate that the TS is better described as a proton transfer. Indeed, inspection of the spin densities of **1**, the H₂-adduct **A**, and the TS **B** reveal little spin-depletion at N (−0.12 for **1**, −0.13 for **A**, −0.15 for **B**) and that the majority of spin density is at uranium (1.19 for **1**, 1.18 for **A**, and 1.24 for **B**) so N-radical character does not appear to play a significant role in the H₂-activation. Following the intrinsic reaction coordinate yields a uranium(V)-imido-hydride complex (**C**), whose formation is almost athermic (loosely endothermic by 2.0 kcal mol⁻¹). Complex **C** can rearrange through a H-atom migration from uranium to parent imido group (transition state **D**), i.e. undergoing a 1,1-migratory insertion, with a reduction of

uranium oxidation state at this point from V to III. The associated activation barrier is 32.1 kcal mol⁻¹ from **C** (34.1 from the start point). The height of this barrier is due to the need of the hydride to be transferred as a proton to the nucleophilic imido group. However, this barrier is kinetically accessible and in-line with the slow reaction observed experimentally. This TS yields trivalent **6** that is thermodynamically stable (−21.0 kcal mol⁻¹). However, in the presence of BMes₃, complex **6** can be easily oxidised into tetravalent **4** in a process that can be considered to

be an essentially athermic electron transfer process since the computed energy difference between **4** and **6** is within the error of the calculation.

In the presence of BMes₃, Fig. 10, the computed reaction pathway is quite different. After the formation of a loosely bonded H₂ adduct, the system reaches an H₂ activation TS, that is reminiscent of FLP complex reactivity. Indeed, at TS **2B**, the H₂ molecule interacts in a bridging end-on fashion with the nitride (that is the nucleophile of the FLP) and the borane (that is the electrophile). Unlike TS **2B**, the H₂ molecule is very little activated at **2A** (1.02 vs 0.83 Å, respectively), and neither the N–H bond (1.68 Å) nor the B–H one (1.69 Å) are yet fully formed. The U–N₍nitride₎ bond distance is similar to that found for **2A** (1.81 Å). The associated barrier is relatively low (14.6 kcal mol⁻¹ with respect to the start point) and similar to the σ-bond metathesis mechanism. Therefore, the presence of BMes₃ does not impact the protonation of the strongly nucleophilic nitride that is very reactive. Again, there is essentially no spin-depletion at the nitride (−0.12 for **1**, −0.13 for **2A**, −0.12 for **2B**) and the unpaired spin density is clearly localised at uranium (1.19 for **1**, 1.20 for **2A**, 1.21 for **2B**), which argues against nitride radical character in this reactivity. The FLP TS **2B** evolves to the formation of a fully dissociative ion pair whose formation is exothermic (−13.9 kcal mol⁻¹ from start point). From the uranium(V)-imido complex **2C**, the formation of trivalent **6** then tetravalent **4** was considered. The first, shown by the gray pathway, implies that the hydroborate (HBMes₃)⁻ delivers the hydrogen to the imido (**2D₁**). However, this route is not favoured because, like the problem in the absence of BMes₃, the hydride has to be transferred as a proton. The computed barrier of 40.4 kcal mol⁻¹ from **2C** (26.5 kcal mol⁻¹ from the start point) is in-line with this. The second possibility, shown by the black pathway, involves a second FLP-type activation of H₂ (**2D₂**). The associated barrier is 10.5 kcal mol⁻¹ lower than **2D₁**, demonstrating the beneficial role of BMes₃. However, the **2D₂**

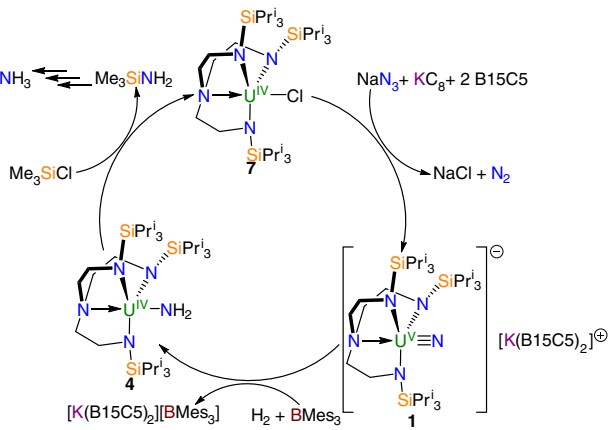

**Fig. 8 Reaction cycle for the production of ammonia.** Treatment of **7** with sodium azide, KC₈, and two equivalents of B15C5 produces the terminal uranium-nitride **1**, which in turn reacts with H₂ and BMes₃ to give **4**. Treatment of **4** with Me₃SiCl, followed by work-up and acidification steps, as indicated by the multiple arrows, produces ammonia. B15C5 = benzo-15-crown-5 ether. Mes = 2,4,6-trimethylphenyl.

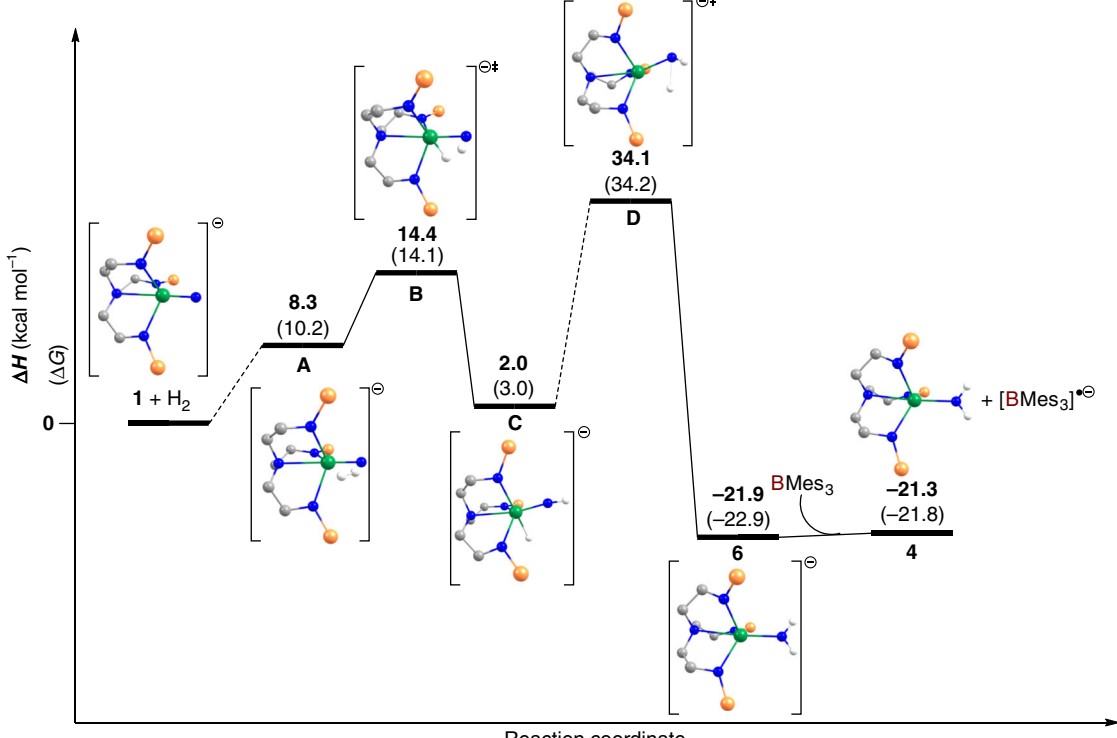

**Fig. 9 Computed reaction profile for the conversion of 1 to 6 in the absence of BMes₃ and then conversion to 4 with the addition of BMes₃.** The iso-propyl groups of the silyl substituents, carbon-bound hydrogen atoms, and [K(B15C5₂)]⁺ cation accommodated in the calculations are omitted for clarity. Bold numbers without parentheses refer to ΔH values and numbers in parentheses are ΔG values, both quoted in kcal mol⁻¹.

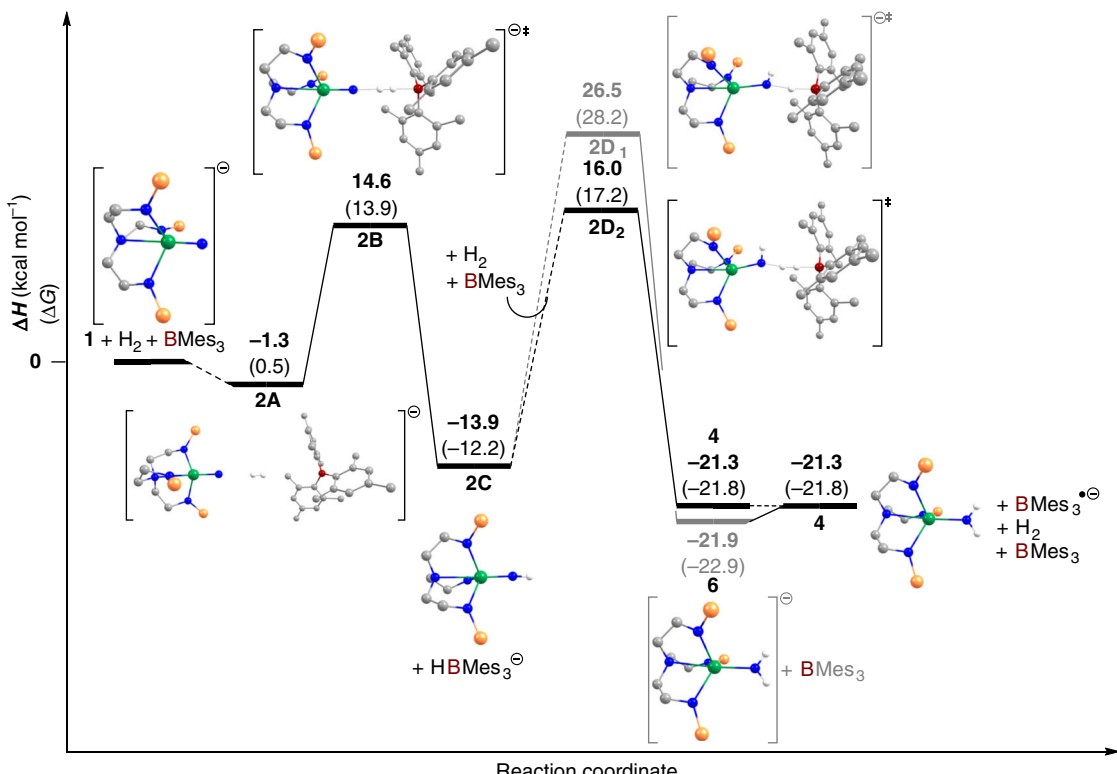

**Fig. 10 Computed reaction profile for the conversion of 1 to 6 and then 4 in the presence of BMes₃.** The *iso*-propyl groups of the silyl substituents, carbon-bound hydrogen atoms, and [K(B15C5₂)]⁺ cation accommodated in the calculations are omitted for clarity. Bold numbers without parentheses refer to $\Delta H$ values and numbers in parentheses are $\Delta G$ values, both quoted in kcal mol⁻¹.

barrier is also higher than the first FLP barrier, indicating that the uranium(V)-imido complex **2C** is a less strong nucleophile than the uranium(V)-nitride **1**. This also evidenced by the geometry at the TS, where the N–H distance is far shorter than in **2B** (1.44 Å vs. 1.68 Å) inducing a shorter B–H distance (1.54 Å vs. 1.69 Å). The resulting more compact geometry enhances steric repulsion that increases the activation barrier. The **2D₂** TS yields ultimately tetravalent **4** (via trivalent **6**) whose formation is exothermic by 21.3 kcal mol⁻¹.

### Discussion

Despite exhaustive attempts, we find no evidence for any reactivity between the uranium(VI)-nitride **2** and H₂ irrespective of whether borane promoters are present or not. However, this is not surprising since prior computational studies have suggested that the U≡N triple bond is rather covalent[37], possibly even more so than group 6 congeners, and so it conforms to the general phenomenon that many metal-nitrides, and especially high oxidation state electron-poor ones, are exceedingly unreactive. To date, CO, CO₂, and CS₂ have been found to react with **2**[40,41], but always much more slowly than **1**, and these small, polar molecules with relatively low-lying π*-orbitals are considerably easier to activate than apolar H₂ that has only a σ*-orbital available for activation.

In contrast, the reactivity of **1** with H₂, the mechanisms of which sharply diverge with or without BMes₃, is surprising and notable because the 5$f^1$ uranium(V) ion in **1** is high oxidation state and cannot be considered to be electron-rich nor low-coordinate. Indeed, the only example of any molecular uranium-nitride reacting with H₂ is the diuranium(IV)-nitride-cesium complex [Cs{U(OSi[OBuᵗ]₃)₃}₂(μ-N)];[47] here, the product is the bridging parent imido-hydride complex [Cs{U(OSi[OBuᵗ]₃)₃}₂(μ-NH)(μ-H)] and this transformation is enabled by the bridging,

polar nature of the nitride and polymetallic cooperativity effects. However, that chemistry stops at the imido-hydride stage, or reverts to nitride and H₂, and does not proceed to the H-atom 1,1-migratory insertion stage to give an amide. When terminal M≡N triple bonds have been found to react with H₂ it is with 4$d^4$ Ru(IV)[26] or 5$d^4$ Os(IV)[27] to give NH₃ and 5$d^6$ Ir(III)[28] to give Ir–NH₂, since these are the only nitrides that are low-coordinate and sufficiently activated and electron-rich enough to reduce the M≡N triple bond orders by populating anti-bonding interactions. This electron-rich activation is not applicable to **1** being only 5$f^1$ and that $f$-electron is in principle nonbonding. However, the nitride is a very strong donor ligand that we have previously shown can modulate the $m_J$ groundstate of uranium depending how strongly it can donate. Specifically, the nitride forms σ- and π-bonds with the $l = 0$ and $l = 1$ 5$f$-orbitals, but also interacts with the $l = 2$ and $l = 3$ 5$f$-orbitals where the 5$f$-electron must reside and thus the supposedly nonbonding 5$f^1$ electron would seem to be not entirely innocent in this circumstance due to an inevitable anti-bonding interaction[38]. Nevertheless, that a metal-nitride of oxidation state as high as +5 and valence electron number as low as one is capable of activating H₂ without utilising ancillary ligand reactivity and H-atom shuttling is unprecedented[1]. Since the reaction profile calculations do not support the notion of nitride radical character promoting the observed and unexpected reactivities, we suggest that this is due to a combination of the 5$f$-electron in **1** not being wholly nonbonding, and also that the uranium(V)-nitride bond is actually more polar than transition metal analogues.

The reaction profile calculations combined with experimental observations provide an internally consistent account of the reactivity reported here. It is clear from experiment that **1** does not bind BMes₃, unlike BPh₃, presumably on steric grounds presenting the potential for FLP chemistry that is intuitively

invoked when considering the steric demands of Tren$^{TIPS}$ and BMes$_3$. When **1** is reacted with H$_2$ in the presence of BMes$_3$ the [U(Tren$^{TIPS}$)(N)]$^-$ and BMes$_3$ components constitute the FLP that can form an encounter complex with H$_2$ and the computed intermediate **2A** and TS **2B** are clear evidence for a FLP encounter complex, which facilitates the splitting of H$_2$, confirming bona fide FLP reactivity and introducing actinide chemistry to the pantheon of FLP reactivity. Although the conversion of **2C** and (HBMes$_3$)$^-$ to **6** and BMes$_3$ is thermodynamically favourable, it is kinetically the least feasible route to occur due to the inherent barrier of a hydride being a proton source, and instead it appears that a second FLP activation of H$_2$ occurs along with oxidation of **6** to give **4**, which is thermodynamically little different to the previous outcome but kinetically more accessible. Within the error of the calculation the oxidation of **6** to **4** is essentially athermic and likely driven by the strongly reducing nature of the uranium(III) ion in **6** coupled to its electron-rich nature. The two-electron reduction on going from **1** to **6** is entirely consistent with the two-electron redox chemistry of H$_2$, and indeed the one-electron oxidation of **6** to **4** is simply a sacrificial one-electron reduction of BMes$_3$ to BMes$_3^{-\bullet}$.

The importance of two FLP reaction steps in the conversion of **1** to **6** and then **4** underscores the importance of the facilitating role that FLP chemistry plays in the hydrogenation of **1**. However, more remarkable is that fact that the FLP component is actually not mandatory for hydrogenolysis of the U≡N triple bond to occur, though its absence does slow the reaction significantly demonstrating the facilitating role of the FLP mechanism since the main origin of this impediment is that formally a proton has to evolve from a hydride. In the absence of an FLP mechanism H$_2$ undergoes a direct 1,2-addition across the U$^V$≡N triple bond to give a H−U$^V$=N−H unit that is reminiscent of the aforementioned reactivity of [Cs{U(OSi[OBu$^t$]$_3$)$_3$}$_2$(μ-N)][47] when their terminal vs bridging natures, respectively, are taken into account. The reactivity of **1** is also reminiscent of aspects of recently reported mechanistic studies of the reaction of uranium(III) with water[66], and it is germane to note that concerted two-electron redox chemistry at uranium remains a relatively rare phenomena[29,36,67] with one-electron processes dominating. The 1,2-addition at **1** is effectively H–H heterolysis to generate H$^+$ and H$^-$, consistent with the polarising nature of the U≡N triple bond. Interestingly, the production of the final U$^{III}$-NH$_2$ linkage in **6** from pentavalent **1** by H-atom 1,1-migratory insertion, consistent with the two-electron reducing nature of H$_2$ since nucleophilic nitrides tend to react without changing metal oxidation state, is reminiscent of the reactivity of uranium(VI)-nitrides under photolytic conditions, where by a R$_3$CH/U$^{VI}$N combination, via a R$_3$C•/U$^V$=N–H intermediate converts to U$^{IV}$-N(H)CR$_3$, since both involve two-electron reductions at uranium overall[37,42,48]. The reactivity of **1** with H$_2$ has parallels to the reactivity of the ruthenium(IV)-nitride complex [Ru{N(CH$_2$CH$_2$PBu$^t$)$_2$}(N)] with H$_2$ to give NH$_3$[26], but with some important differences. The Ru-complex initially reacts with H$_2$ across the Ru-N$_{amide}$ not Ru-N$_{nitride}$ bond, so like many nitrides when reactivity occurs it is with the ancillary ligand not the metal-nitride linkage itself as is the case with **1**. However, the Ru-complex does at a later stage transfer a H-atom from Ru to an imido group to form a Ru-NH$_2$ group like **C/2C**. In contrast, the iridium(III)-nitride complex [Ir{NC$_5$H$_3$-2,2′-(C[Me] = N-2,6-Pr$^i_2$C$_6$H$_3$)$_2$}(N)] is reported to undergo concerted reactivity with H$_2$ to directly afford an amide and no prior coordination of H$_2$ to the Ir centre[28]. Looking more widely to sulfido chemistry, the complex [Ti(η$^5$-C$_5$Me$_5$)$_2$(S)(NC$_5$H$_5$)] reacts with H$_2$ to give the hydrosulfide-hydride [Ti(η$^5$-C$_5$Me$_5$)$_2$(SH)(H)][68,69], providing a parallel to the 1,2-addition of H$_2$ across the U≡N triple bond of **1**,

but unlike **1** the titanocene reactivity halts at the hydrosulfide-hydride formulation and does not undergo a subsequent H-atom 1,1-migratory insertion since that would require formation of SH$_2$ and formally the unfavourable reduction of titanium(IV) to titanium(II). So, the reactivity of **1** displays similar and divergent reactivity pathways to known transition metal-nitride reactivity, but combines 1,2-addition and 1,1-migratory insertion steps where transition metals tend to execute either 1,2-additions or 1,1-insertions at the M≡E bond, but are not capable of executing both together.

The reactivity of **4** with BPh$_3$ and H$_2$ is notable, though complex because it would seem products react to give reactants, because again it invokes the notion of FLP chemistry whereby weakly coordinated [U(Tren$^{TIPS}$)]$^+$ and [H$_2$NBPh$_3$]$^-$ components are sufficiently activated to cleave H$_2$ to give H$_3$NBPh$_3$. While this is currently of no practical use it demonstrates the potential for further FLP hydrogenolysis chemistry to convert the parent amide to ammonia. However, we have demonstrated a reaction cycle, where azide is converted to nitride, which undergoes hydrogenolysis to amide, and the amide can be quenched by acid to give ammonia. Thus, overall a nitride has been hydrogenated to ammonia, and the experimentally and computationally supported proposed reactivity mechanisms contribute to our wider understanding of the reactivity of uranium-nitrides toward H$_2$ in heterogeneous Haber Bosch and ATF scenarios.

In summary, while the uranium(VI)-nitride **2** is apparently inert with respect to reacting to H$_2$, the uranium(V)-nitride **1** is not, suggesting that the 5$f$-electron of the latter is not entirely nonbonding and that the nitride imposes a strong ligand field on uranium. The absence of reactivity for **2** is entirely in-line with the lack of reactivity for high oxidation state metal-nitrides generally, but the latter is not and is notable for being neither low-coordinate nor electron-rich, which are the two requirements previously common to all terminal metal-nitrides that react with H$_2$, yet it is reactive. This study reveals two distinct H$_2$-activation mechanisms. When the borane BMes$_3$ (Mes = 2,4,6-trimethylphenyl) is present a FLP mechanism operates where two H$_2$ heterolysis events and a borane reduction step sequentially combine to furnish a U$^{IV}$-NH$_2$ product, and this, to the best of our knowledge, is the first demonstration of the application of bona fide FLP reactivity to actinide chemistry. When the borane is absent, direct 1,2-addition of H$_2$ across the U≡N triple bond to give a H−U$^V$=N−H intermediate followed by H-atom migration produces a U$^{III}$−NH$_2$ product that is easily oxidised to U$^{IV}$−NH$_2$. The direct hydrogenolysis addition is slower than the FLP-mediated mechanism, demonstrating the facilitating role of FLPs. We find evidence that treating the U$^{IV}$−NH$_2$ product with BPh$_3$ and H$_2$ produces further FLP hydrogenolysis reactivity, since H$_3$NBPh$_3$ has been detected in reaction mixtures, but this is reversible and produces products that react to give the starting materials. We have demonstrated an azide to nitride to amide to ammonia reaction cycle, supported by overall hydrogenation involving hydrogenolysis and electrophilic quenching steps. Thus, overall a nitride has been converted to ammonia, and the experimentally and computationally supported proposed reactivity mechanisms inform our understanding of the reactivity of uranium-nitrides towards H$_2$ in heterogeneous Haber Bosch and ATF scenarios.

## Methods

**General**. Experiments were carried out under a dry, oxygen-free dinitrogen atmosphere using Schlenk-line and glove-box techniques. All solvents and reagents were rigorously dried and deoxygenated before use. Compounds were variously characterised by elemental analyses, NMR, FTIR, EPR, and UV/Vis/NIR electronic

absorption spectroscopies, single crystal X-ray diffraction studies, Evans and SQUID magnetometry methods, and DFT computational methods.

**Preparation of [U(Tren$^{TIPS}$)(NBPh$_3$)][K(B15C5)$_2$] (3).** Toluene (20 ml) was added to a stirring mixture of **1** (0.54 g, 0.37 mmol) and BPh$_3$ (0.09 g, 0.37 mmol). The resulting mixture was stirred for 16 h to afford a brown precipitate. The mixture was briefly heated to reflux and filtered. Volatiles were removed in vacuo. The resulting brown solid subsequently identified as **3** was washed with pentane (3 × 5 ml) and dried in vacuo. Yield of **3**: 0.42 g, 66%. X-ray quality crystals were grown in benzene solution at room temperature. Anal. calcd for C$_{79}$H$_{130}$BKN$_5$O$_{10}$Si$_3$U: C, 56.41; H, 7.79; N, 4.16%. Found: C, 56.48; H, 7.82; N, 3.95%. $^1$H NMR (C$_6$D$_6$, 298 K): δ 39.15 (s, 6 H, CH$_2$), 23.53 (s, 6H, CH$_2$), 10.69 (s, 6H, Ar-*H*), 9.27 (s, 3H, Ar-*H*), 7.26–6.96 (br m, 14H, Ar-*H*), 4.47–4.20 (m, 32H, OCH$_2$), −7.92 (s, 54H, Pr$^i$-CH$_3$), −9.12 (s, 9H, Pr$^i$-C*H*). $^{11}$B{$^1$H} NMR (C$_6$D$_6$, 298 K): δ 112.8. FTIR: υ/cm$^{-1}$: 2938 (w), 2914 (w), 2859 (m), 1592 (m), 1503 (m), 1454 (m), 1427 (w), 1405 (w), 1382 (w), 1360 (w), 1331 (w), 1288 (m), 1254 (m), 1217 (m), 1125 (s), 1097 (m), 1064 (m), 1050 (m), 1046 (m), 1009 (w), 993 (w), 934 (s), 881 (m), 852 (m), 836 (m), 795 (m), 781 (m), 721 (s), 703 (s), 664 (m), 640 (m), 609 (m), 566 (m), 550 (m), 511 (w), 501 (w), 465 (m), 448 (m). μ$_{eff}$ (Evans method, C$_6$D$_6$, 298 K): 1.96 μ$_B$.

**Attempted reaction of [U(Tren$^{TIPS}$)(N)][K(B15C5)$_2$] (1) with H$_2$ and BPh$_3$.** A brown solution of **1** (0.040 g, 0.03 mmol) and BPh$_3$ (0.007 g, 0.03 mmol) in C$_6$D$_6$ (0.5 ml) was degassed and exposed to an atmosphere of H$_2$. The brown solution was analysed for 7 days by $^1$H NMR spectroscopy, after which only the formation of **3** was observed.

**Attempted reaction of [U(Tren$^{TIPS}$)(N)][K(B15C5)$_2$] (1) with BMes$_3$.** A colourless solution of BMes$_3$ (0.01 g, 0.03 mmol) in C$_6$D$_6$ (0.5 ml) was added to **1** (0.04 g, 0.03 mmol). The brown solution was analysed $^1$H NMR spectroscopy, revealing resonances of free BMes$_3$ and **1**. $^1$H NMR (C$_6$D$_6$, 298 K): δ 38.58 (s, 6H, CH$_2$), 16.98 (s, 6H, CH$_2$), 10.03–6.80 (br m, 20H, CH$_2$, OCH$_2$, Ar–*H*), 6.72 (s, 6H, Ar-*H*$_{BMes3}$), 3.85 (s, 20H, OCH$_2$), 2.16 (s, 12H, CH$_{3BMes3}$), 2.14 (s, 9H, CH$_{3BMes3}$), −5.61 (s, 9H, Pr$^i$-C*H*), −6.30 (s, 54H, Pr$^i$-CH$_3$). $^{11}$B{$^1$H} NMR (C$_6$D$_6$, 298 K): δ 76.76.

**Reaction of [U(Tren$^{TIPS}$)(N)][K(B15C5)$_2$] (1) with H$_2$ and BMes$_3$.** A brown solution of **1** (0.40 g, 0.28 mmol) and BMes$_3$ (0.10 g, 0.72 mmol) in toluene (20 ml) was degassed and exposed to H$_2$ (1 atm.). The mixture was stirred for 2 days to ensure the complete consumption of the starting material and **5** started to precipitate as a dark blue solid after 1 day of stirring. The supernatant of **5** was removed by filtration. Dark blue **5** was washed with toluene (3 × 10 ml) and dried in vacuo. Yield of **5**: 0.18 g, 69%. The volatiles of the supernatant were removed in vacuo yielding an oily brown residue containing **4** as the main uranium product. Yield of **4**, based on $^1$H NMR spectroscopy: 67%. $^1$H NMR of **4** (C$_6$D$_6$, 298 K): δ 107 (s, 2H, NH$_2$), 31.99 (s, 6H, CH$_2$), 7.92 (s, 6H, CH$_2$), −5.35 (s, 9H, Pr$^i$–C*H*), −5.87 (s, 54H, Pr$^i$–CH$_3$). A similar reaction using D$_2$ instead of H$_2$ leads to the formation of **4/4′/4″** (**4′**, $^1$H δ 106 ppm, $^2$H δ 106.8 ppm, $^2$J$_{HD}$ not resolved; **4″**, $^1$H δ no resonance in the 100–110 ppm region, $^2$H δ 107.5 ppm). Ammonia liberation after treatment of **4/4′/4″** with 1 equivalent of HCl led the formation of NH$_3$DCl. [B(Mes)$_3$][K(B15C5)$_2$] (**5**): Anal. calcd for C$_{55}$H$_{73}$KO$_{10}$: C, 69.97; H, 7.79; N, 0%. Found: C, 69.81; H, 7.88; N, 0%. FTIR υ/cm$^{-1}$: 2906 (w), 2873 (w), 1582 (w), 1503 (m), 1452 (m), 1362 (w), 1331 (w), 1291 (w), 1252 (m), 1240 (m), 1219 (m), 1184 (w), 1123 (m), 1099 (m), 1074 (m), 1044 (m), 1005 (m), 936 (m), 852 (m), 840 (m), 813 (w), 777 (w), 748 (m), 734 (m), 695 (w), 675 (w), 603 (w), 573 (w), 562 (w), 542 (w), 509 (w), 467 (m), 454 (w), 411 (w).

**Reaction of [U(Tren$^{TIPS}$)(N)][K(B15C5)$_2$] (1) with H$_2$ or D$_2$.** With *H$_2$*: a brown solution of **1** (1.16 g, 0.81 mmol) in toluene (20 ml) was degassed and exposed to H$_2$ (1 atm.). The mixture was stirred for 7 days at 15 °C to ensure the complete consumption of the starting material and formation of **6** as a grey solid. The supernatant of the grey solid was removed by filtration. The solid was washed with toluene (3 × 10 ml) and dried in vacuo. Yield of **6**: 0.53 g, 45%. The volatiles of the filtrate were removed in vacuo yielding an oily brown residue containing traces of **4**, B15C5, and Tren$^{TIPS}$H$_3$. With *D$_2$*: a brown solution of **1** (1.03 g, 0.71 mmol) in toluene (20 ml) was degassed and exposed D$_2$ (1 atm.). The mixture was stirred for 7 days at 15 °C to ensure complete consumption of the starting material and formation of **6/6′/6″** as a grey solid. The supernatant of the grey solid was removed by filtration. The solid was washed with toluene (3 × 10 ml) and dried in vacuo. Yield of **6/6′/6″**: 0.68 g, 66%. The volatiles of the filtrate were removed in vacuo yielding an oily brown residue containing traces of **4/4′/4″**, B15C5, and Tren$^{TIPS}$H$_3$. X-ray quality crystals of **6** were grown from a 0.069 g/ml solution of **1** in toluene exposed to an atmosphere of H$_2$ for three weeks. Anal. calcd for C$_{61}$H$_{117}$KN$_5$O$_{10}$Si$_3$U: C, 50.81; H, 8.18; N, 4.86%. Found: C, 50.64; H, 8.38; N, 4.95%. FTIR υ/cm$^{-1}$: 2940 (m), 2916 (w), 2881 (w), 2851 (m), 2814 (w), 1596 (w), 1505 (m), 1456 (m), 1411 (w), 1364 (w), 1348 (w), 1333 (w), 1295 (w), 1272 (w), 1254 (m), 1219 (m), 1125 (s), 1107 (m), 1097 (m), 1078 (m), 1046 (m), 1007 (w), 983 (w), 936 (s), 883 (m), 854 (m), 799 (w), 775 (w), 746 (s), 671 (m, for H$_2$ only),

664 (m), 654 (w), 622 (m), 603 (w), 591 (w), 560 (w), shoulder 544 (w, for D$_2$ only), 536 (w), 530 (w), 505 (w), 467 (w), 458 (w), 440 (m), 424 (w). The insolubility of **6** once isolated precluded the determination of its $^1$H NMR spectrum, the solution magnetic moment by Evans method, and acquisition of a UV/Vis/NIR electronic absorption spectrum. Heating a suspension of **6** in C$_6$D$_6$ resulted in the observation of resonances that correspond to **4** as evidenced by $^1$H NMR spectroscopy.

**Reaction between [U(Tren$^{TIPS}$)(NH$_2$)][K(B15C5)$_2$] (6) and BMes$_3$.** A colourless solution of BMes$_3$ (0.01 g, 0.03 mmol) in 0.5 ml of C$_6$D$_6$ was added to **6** (0.04 g, 0.03 mmol) resulting to the formation of an intense blue solution characteristic of the formation of the radical anion BMes$_3$$^{•−}$. Rapidly, dark blue crystals of **5** formed and $^1$H NMR spectrum revealed the formation of **4** in 52% yield.

**Reaction of [UN(Tren$^{TIPS}$)][K(B15C5)$_2$] (1) with 9,10-dihydroanthracene.** A J Youngs-valve NMR tube was charged with **1** (36 mg, 25 μmol) and 9,10-dihy-droanthracene (4.5 mg, 25 μmol). C$_6$D$_6$ (0.8 ml) was added and the resulting brown mixture was left to stand. After 10 min a turbid red mixture was observed. After standing for 24 h the resulting brown mixture was analysed by $^1$H and $^2$H NMR spectroscopy with the only observable uranium containing product being **4**. During that time a small amount of red crystalline material deposited that was identified as [K(B15C5)$_2$][C$_{14}$H$_{11}$] by a combination of X-ray diffraction and $^1$H NMR spectroscopy when redissolved. $^1$H NMR (C$_4$D$_8$O, 298 K): δ 3.59–3.64, 3.67–3.91, 3.88–3.92 (br, m, 32H, OCH$_2$), 4.41 (s, C=C*H*), 5.62 (td, 2H, 1,9-Anth-C*H*, $^3$J$_{HH}$ = 6.85 Hz, $^3$J$_{HH}$ = 1.22 Hz), 5.89 (dd, 2H, 4,6-Anth-CH, $^3$J$_{HH}$ = 8.31 Hz, $^3$J$_{HH}$ = 1.22 Hz), 6.25 (t, $^3$J$_{HH}$ = 6.60 Hz, 2,3,7,8-Anth-C*H*), 6.75–6.85 (br, m, 8H, OCHC*H*). Resonances for the CH$_2$ group were not observed and are likely obscured by residual $d_8$-THF or crown ether resonances between 3.55 and 3.92 ppm.

**Ammonia formation after addition of 1 equivalent of HCl to [U(Tren$^{TIPS}$) (NHD)] (4′).** Complex **4′** (0.05 g, 0.06 mmol), formed from the reaction of **1** with D$_2$ in the presence of BMes$_3$, was treated with 1.2 ml of a 0.05 M HCl solution in THF/Et$_2$O (0.06 mmol) and stirred for 2 h at room temperature. All volatiles were then vacuum transferred onto a 2 M HCl solution in Et$_2$O (2 ml). Volatiles were removed in vacuo and the resulting white solid was dissolved in 0.6 ml of $d_6$-DMSO to quantify the amount of ammonia present using $^1$H NMR spectroscopy (quantification using sealed capillary insert of 2,5-dimethylfuran in $d_6$-DMSO)[59]. Integration of the NH$_3$$^+$ multiplet (7.30 ppm) revealed 40% NH$_3$DCl. The $^2$H NMR spectrum revealed the presence of a broad resonance at 7.12 ppm. Addition of 10 μl of H$_2$O gave complete proton/deuterium exchange, as the resonance at 7.12 ppm in the $^2$H NMR spectrum disappeared and a NH$_4$$^+$ 1:1:1 triplet (7.28 ppm, $J_{NH}$ = 51 Hz) was formed, integration of the triplet revealed 52% NH$_4$Cl.

**Ammonia formation after addition of 1 equivalent of HCl to [U(Tren$^{TIPS}$) (NHD)][K(B15C5)$_2$] (6′).** Complex **6′** (0.03 g, 0.021 mmol) was treated with 2.1 ml of a 0.01 M HCl solution in THF/Et$_2$O (0.02 mmol) and was stirred for 2 h at room temperature. All the volatiles were then vacuum transferred into a 2 M HCl solution in Et$_2$O (2 ml). Volatiles were removed in vacuo and the resulting white solid was dissolved in 0.6 ml of $d_6$-DMSO to quantify the amount of ammonia present using $^1$H NMR spectroscopy (quantification using sealed capillary insert of 2,5-dimethyl-furan in $d_6$-DMSO)[59]. Analysis of the brown solid residue after distillation of the volatiles revealed the presence of **7** in 1–5% yield with Tren$^{TIPS}$H$_3$ as main product. Integration of the NH$_3$D$^+$ multiplet (7.30 ppm) revealed 35% NH$_3$DCl. The $^2$H NMR spectrum revealed the presence of a broad resonance at 7.12 ppm. Addition of 10 μl of H$_2$O gave complete proton/deuterium exchange, as the resonance at 7.12 ppm in the $^2$H NMR spectrum disappeared and a NH$_4$$^+$ 1:1:1 triplet (7.28 ppm, $J_{NH}$ = 51 Hz) was formed, integration of the triplet revealed 46% NH$_4$Cl.

**Ammonia formation after addition of 1 equivalent of DCl to [U(Tren$^{TIPS}$) (NHD)][K(B15C5)$_2$] (6′).** Complex **6′** (0.04 g, 0.03 mmol) was treated with 2.8 ml of a 0.01 M DCl solution in THF/Et$_2$O (0.03 mmol) and stirred for 2 h at room temperature. All volatiles were then vacuum transferred into a 2 M HCl solution in Et$_2$O (2 ml). Volatiles were removed in vacuo and the resulting white solid was dissolved in 0.6 ml of $d_6$-DMSO to quantify the amount of ammonia present using $^1$H NMR spectroscopy (quantification using sealed capillary insert of 2,5-dimethylfuran in $d_6$-DMSO)[59]. Integration of the NHD$_3$$^+$ multiplet (7.37 ppm) revealed 11% NHD$_3$Cl. The $^2$H NMR spectrum revealed the presence of a broad triplet at 7.24 ppm. Addition of 10 μl of H$_2$O gave complete proton/deuterium exchange, as the resonance at 7.24 ppm in the $^2$H NMR spectrum disappeared and a NH$_4$$^+$ 1:1:1 triplet (7.28 ppm, $J_{NH}$ = 51 Hz) was formed, integration of the triplet revealed 48% NH$_4$Cl.

**Synthesis of [U(Tren$^{TIPS}$)(NH$_2$BPh$_3$)] (8).** Toluene (20 ml) was added to a stirring mixture of **4** (0.20 g, 0.23 mmol) and BPh$_3$ (0.06 g, 0.23 mmol). The resulting mixture was stirred for a further 16 h to afford a brown precipitate. The mixture was filtered and volatiles were removed in vacuo. X-ray quality crystals grew in the brown oily residue overnight. Crystals were washed with pentane (2 ×

5 ml) and dried in vacuo. Yield of **8**: 0.16 g, 62%. Anal. calcd for $C_{51}H_{92}BN_5Si_3U$: C, 55.26; H, 8.37; N, 6.32%. Found: C, 55.52; H, 8.16; N, 5.91%. NMR spectroscopy reveals that when isolated **8** is dissolved in solution it dissociates to **4** and free $BPh_3$ and also trace $H_3NBPh_3$ and **9**, but this equilibrium can be manipulated by cooling samples favouring the formation of **8** so a variable-temperature NMR study was performed, see below. The presence of $H_3NBPh_3$ could not be unequivocally confirmed in the $^1H$ NMR spectrum due to its low concentration level in a spectrum dominated by paramagnetic species, but its presence is confirmed by $^{11}B$ NMR spectroscopy. Trace resonances corresponding to reported data for **9** could be observed[65]. $^{11}B\{^1H\}$ NMR ($C_6D_6$, 298 K): $\delta$ 67.5 ($BPh_3$), −3.20 ($H_3NBPh_3$), −55.2 (U-$H_2NBPh_3$). FTIR: $\upsilon/cm^{-1}$: 3293 (w), 3228 (w), 3044 (w), 2938 (m), 2886 (m), 2861 (m), 1590 (w), 1502 (w), 1461 (m), 1428 (m), 1372 (m), 1339 (w), 1316 (w), 1270 (m), 1238 (m), 1166 (w), 1133 (w), 1116 (w), 1047 (m), 1010 (m), 988 (m), 925 (s), 880 (s), 816 (w), 731 (s), 701 (s), 670 (s), 632 (s), 596 (m), 565 (m), 554 (m), 514 (s). $\mu_{eff}$ (Evans method, $C_6D_6$, 298 K): 2.96 $\mu_B$.

**Variable-Temperature NMR study of 8.** A brown solution of **4** (0.04 g, 0.05 mmol) in $d_8$-toluene (0.3 ml) was added to $BPh_3$ (0.01 g, 0.05 mmol) in $d_8$-toluene (0.2 ml). The brownish black solution was analysed by $^1H$ and $^{11}B\{^1H\}$ NMR spectroscopies at 293, 253, and 233 K. Integrations are listed relatively for functional units within a given species, but note at 293 K **8** is fully dissociated to **4**, at 253 K the ratio of **4**:**8** is ~2:1, and at 233 K that ratio is then ~2:3. $^1H$ NMR ($C_6D_5CD_3$, 293 K): $\delta$ 107 (s, 2H, $NH_2$, **4**), 31.99 (s, 6H, $CH_2$, **4**), 7.92 (s, 6H, $CH_2$, **4**), 7.4–6.2 (m, br, 15H, $B(C_6H_5)_3$), −5.35 (s, 9H, $Pr^i$-$CH$, **4**), −5.87 (s, 54H, $Pr^i$-$CH_3$, **4**). $^{11}B\{^1H\}$ NMR ($C_6D_5CD_3$, 298 K): $\delta$ 67.5 ($BPh_3$), −3.2 ($H_3NBPh_3$). $^1H$ NMR ($C_6D_5CD_3$, 253 K): $\delta$ 148.1 (s, br, 2H, $NH_2$, **4**), 43.5 (s, br, 6H, $CH_2$, **4**), 11.5 (s, vbr, 63H, $Pr^i$-$CH$ and $Pr^i$-$CH_3$, **8**), 9.2 (s, br, 6H, $CH_2$, **4**), 7.5–3.7 (s, vbr, 15H, $B(C_6H_5)_3$), −7.5 (s, br, 9H, $Pr^i$-$CH$, **4**), −8.5 (s, br, 54H, $Pr^i$-$CH_3$, **4**), −34.5 (s, vbr, 6H, $CH_2$, **4**), −44.5 (s, vbr, 6H, $CH_2$, **8**), −154.4 (s, vbr, 1H, $NH_2$, **8**), −173.6 (s, vbr, 1H, $NH_2$, **8**). $^{11}B\{^1H\}$ NMR ($C_6D_5CD_3$, 253 K): $\delta$ 69.0 ($BPh_3$), −95.8 (U-$H_2NBPh_3$, **8**). $^1H$ NMR ($C_6D_5CD_3$, 233 K): $\delta$ 171.9 (s, br, 2H, $NH_2$, **4**), 50.3 (s, br, 6H, $CH_2$, **4**), 16.7 (s, br, 9H, $Pr^i$-$CH$, **8**), 15.8 (s, br, 6H, $CH_2$, **4**), 10.04 (s, br, 54H, $Pr^i$-$CH_3$, **8**), 3.7–2.1 (m, br, 15H, $H_2NB(C_6H_5)_3$) −8.5 (s, br, 9H, $Pr^i$-$CH$, **4**), −9.5 (s, br, 54H, $Pr^i$-$CH_3$, **4**), −38.4 (s, br, 6H, $CH_2$, **8**), −51.2 (s, br, 6H, $CH_2$, **8**), −159.6 (s, br, 1H, $NH_2$, **8**), −196.1 (s, br, 1H, $NH_2$, **8**). $^{11}B\{^1H\}$ NMR ($C_6D_5CD_3$, 233 K): $\delta$ −107.9 (U-$H_2NBPh_3$, **8**).

**Reaction between [U(Tren$^{TIPS}$)(NH₂)] (4) and Me₃SiCl.** $Me_3SiCl$ (6 μl, 0.05 mmol) was added to a brown solution of **4** (0.04 g, 0.05 mmol) in benzene (0.5 ml). The mixture was stirred at room temperature for 2 h. All volatiles containing N-silylated products were distilled under reduced pressure and stirred for 12 h into an aqueous solution of $H_2SO_4$ (0.5 M, 5 ml) to convert the N-silylated products into ammonium salts[70]. After the addition of an excess amount of base (aqueous 30% KOH, 5 ml), ammonia was distilled into HCl solution in $Et_2O$ (2 M, 2 ml) under reduced pressure. The amount of ammonia was determined by $^1H$ NMR spectroscopy using sealed capillary insert of 2,5-dimethylfuran in $d_6$-DMSO[59]. Yield $NH_3$: 53%. To the residual solid fraction containing uranium complexes was added ferrocene as an internal standard in $C_6D_6$ (0.5 ml) to quantify the amount of **7** formed. Yield of **7**: 46%.

## Data availability

The X-ray crystallographic coordinates for the structures of **3**, **5**, **6**, and **8** reported in this study have been deposited at the Cambridge Crystallographic Data Centre (CCDC), under deposition numbers 1870831–1870834 and 1936479. These data can be obtained free of charge from The Cambridge Crystallographic Data Centre via www.ccdc.cam.ac.uk/data_request/cif. $^1H$ NMR spectroscopic data for **3**, **4**, and **8** can be found in Supplementary Figs. 15–17. All other data can be obtained from the authors on request.

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

## Acknowledgements

We gratefully acknowledge funding and support from the UK Engineering and Physical Sciences Research Council (grants EP/K024000/1, EP/M027015/1, and EP/P001386/1), European Research Council (grant CoG612724), Royal Society (grant UF110005), Humboldt Foundation, CalMip, The National EPSRC UK EPR Facility, The University of Manchester, COST Action CM1006, and the Leverhulme Trust UK for a research fellowship to FT. We thank Prof. D. W. Stephan (University of Toronto) for insightful discussions about this work.

## Author contributions

L.C. prepared the compounds and recorded and interpreted the characterisation data. E.L., I.D., and L.M. conducted the reaction profile calculations and analysed the results. E.L. and F.T. recorded and interpreted the magnetic and EPR spectroscopic data. A.J.W. collected, solved, and refined the X-ray crystallographic data and conducted the experiments to identify the H/D-scrambling source. B.M.G. conducted preliminary experiments with the BCF reagent. S.T.L. originated and developed the central idea, analysed all the data, and wrote the manuscript with input from all authors.

## Competing interests

The authors declare no competing interests.
