## [Peer Review File · Nature Communications]

Editorial Note: This manuscript has been previously reviewed at another journal that is not operating a transparent peer review scheme. This document only contains reviewer comments and rebuttal letters for versions considered at *Nature Communications*. Mentions of the other journal and prior referee reports have been redacted.

Reviewers' comments:

Reviewer #1 (Remarks to the Author):

This manuscript reports ammonia synthesis by terminal U(V)-hydrogenations and two possible reaction mechanisms. The ammonia molecule is formed from a terminal nitrogen, bound to uranium, reacting with hydrogen gas. That is, the N₂ molecule was previously broken by the chemical reactions that yielded the terminal N on U. [REDACTED]

[REDACTED] In my read I had pause on similar issues as those raised previously (more below) as I find that those questions have not been fully addressed in the new version. The experimental data seem thorough and convincing, particularly after the D-labeled studies. However, the theoretical investigation appears incomplete and could be significantly strengthened, as outlined below. Until these key questions are fully addressed, this manuscript is not meritorious of a high-level general-interest journal such as Nat. Comm.

1. Configurations along reaction pathways

Studying reaction mechanism is very challenging as one has to propose and investigate all possible chemical configurations to identify the most plausible reaction pathway. Indeed, the stability of each structure directly impact the reaction energy profile along the reaction pathways. It appears to this reviewer that the energy landscape at each state is not carefully explored in this work. I only illustrated this point with a few examples:

a.) For the configuration A in Figure 9, where H₂ is bound to the metal center, is that the lowest configuration? How does that configuration compare to the configuration bound to the terminal nitrogen, similar to 2B structure in Figure 10? This kind of energy landscape will be useful and could be included in Supporting Information.

b.) For the configuration 2A in Figure 10, it was determined to be a TS but "H₂ molecule is very little activated, and neither the N-H bond nor the B-H one are yet formed". Since no bond is activated, why is it defined as a TS? What mode corresponds to the imaginary frequency? Is it along the reaction coordinate?

c.) In Figure 10, structures are missing from step 2C to 2D by adding H₂ and B(Mes)₃. If the energetics of these steps are higher than the energy of 2D, then the current analysis and conclusions do not stand.

d.) There are other reaction pathways are not explored, one intuitive one is the direct addition of H₂ to terminal nitrogen, etc.

the authors should reconsider their choice of title and modify the wordings about “synthetic cycle” since the emphasis of ammonia synthesis adds little value to the manuscript but causes a lot of confusion and could be misleading to the reviewers and more importantly to the readers.

Major comments:

1) This reviewer doesn't see the point of putting “ammonia synthesis” in the beginning of the title. [REDACTED]

“Hydrogenations of terminal uranium(V)-nitride involving direct-addition or Frustrated Lewis Pair mechanisms” or so is more appropriate.

2) Relevantly, the last sentence in the abstract “A synthetic hydrogenation cycle for ammonia synthesis is demonstrated, and this work establishes a unique marriage of actinide and FLP chemistries.” should be removed or replaced. Hydrogenation by definition means “a chemical reaction between molecular hydrogen (H₂) and another compound or element”. In this report, the hydrogenation of terminal uranium(V) nitride only led to uranium amide but not uranium hydride and free ammonia. A genuine synthetic hydrogenation cycle for ammonia synthesis should start from N₂ and H₂ and end with NH₃. Here, N was from NaN₃; uranium nitride was generated from salt metathesis between U-Cl and NaN₃ in the presence of KC₈; the release of ammonia required treatment of U-NH₂ with Lewis acid Me₃SiCl followed by hydrolysis of Me₃SiNH₂. By all means, this should not be called “a synthetic hydrogenation cycle for ammonia synthesis” since only one step (M-nitride to M-NH₂) was achieved by hydrogenation and the rest involves multiple transformations and N source was NaN₃ but not N₂.

3) The authors claimed that “However, in contrast 1 reacts with H₂ under mild conditions despite the fact it is a high oxidation state metal and not of a low coordination number nor electron-rich as a 5f¹ metal ion.” This is not true. First, U(V) should not be considered high oxidation state metal compared to Ru(IV) and Os(IV). Actually, Ru(IV) and Os(IV) is more oxidizing than U(V) in normal conditions. Secondly, taking account the size of metal ions, U(V) is much larger than Ru(IV) and Os(IV), therefore, the coordination number of 5 in uranium(V) complexes should not be considered “not low coordination number” while the coordination number of 4 is called low coordination number by the authors for Ru(IV) and Os(IV) nitride complexes. Indeed, in the Ru and Os cases, the metal nitride complexes were able to evolve NH₃ but here in the U case, the authors were unable to achieve the hydrogenolysis of U-NH₂. About electron-rich or poor, 5f¹ metal ion is not necessary less electron-rich than a d⁴ metal ion. For instance, Cu(III) is d⁸ metal ion but no one would call it electron-rich; while Th(III) is a 6d¹ metal ion but no one would call it not electron-rich. This all depends on the redox potentials as well as the orbital matching. Here, U(V) is more readily oxidizable and less likely to be reduced than Ru(IV) and Os(IV). Therefore, this reviewer doesn't see any point to call the U(V) in this study “a high oxidation state metal, not low coordination number, not electron-rich” compared to the literature precedents Ru(IV) and Os(IV).

4) This reviewer disagrees with the authors about the reasons that not providing the full NMR spectra for compounds 3,4,5,8. We are in the digital era. In order to improve the reproducibility as well as the transparency of the published research, all possible characterization should be provided upon publication. This reviewer believes that the editors and the authors can work out a way to find a solution to include full ^1H NMR spectra for compounds 3,4,5,8. And if not, a specific reason for not able to obtaining a satisfactory clean NMR spectrum should be provided.

Reviewer Comments:

Reviewer #1:

This manuscript reports ammonia synthesis by terminal U(V)-hydrogenations and two possible reaction mechanisms. The ammonia molecule is formed from a terminal nitrogen, bound to uranium, reacting with hydrogen gas. That is, the N₂ molecule was previously broken by the chemical reactions that yielded the terminal N on U. [REDACTED]

[REDACTED] In my read I had pause on similar issues as those raised previously (more below) as I find that those questions have not been fully addressed in the new version. The experimental data seem thorough and convincing, particularly after the D-labeled studies. However, the theoretical investigation appears incomplete and could be significantly strengthened, as outlined below. Until these key questions are fully addressed, this manuscript is not meritorious of a high-level general-interest journal such as Nat. Comm.

1. Configurations along reaction pathways

Studying reaction mechanism is very challenging as one has to propose and investigate all possible chemical configurations to identify the most plausible reaction pathway. Indeed, the stability of each structure directly impact the reaction energy profile along the reaction pathways. It appears to this reviewer that the energy landscape at each state is not carefully explored in this work. I only illustrated this point with a few examples:

RESPONSE: We absolutely agree that finding the most plausible reaction pathway is not an easy task, but on the other hand Maron has been working in this field for more than twenty years, more than ten years on actinide reactivity, and he is widely acknowledged to be the 'go-to' expert for the entire field where calculating reliable and accurate actinide reaction profiles is concerned.

a.) For the configuration A in Figure 9, where H₂ is bound to the metal center, is that the lowest configuration? How does that configuration compare to the configuration bound to the terminal nitrogen, similar to 2B structure in Figure 10? This kind of energy landscape will be useful and could be included in Supporting Information.

RESPONSE: We are somewhat surprised by such a comment because end-on coordination of H₂ to uranium is completely impossible as there is no lone pair located on H, thus only side on coordination involving donation from the $\sigma(\text{H}_2)$ orbital is observed. Also, back-donation from the metal to the $\sigma^*(\text{H}_2)$ is very small here since the two orbitals have not the same symmetry. If considering end on to the nitride alone, every attempt yielded the TS reported in the manuscript, in line with a σ -bond metathesis reaction of H₂.

b.) For the configuration 2A in Figure 10, it was determined to be a TS but "H₂ molecule is very little activated, and neither the N-H bond nor the B-H one are yet formed". Since no bond is activated, why is it defined as a TS? What mode corresponds to the imaginary frequency? Is it along the reaction coordinate?

RESPONSE: This is a surprising comment because it is not correct to posit that "no bond is activated" since we stated it is "little activated", which means it is activated just not hugely so at that point. Indeed, the analysis of the TS geometry is very informative regarding the height of the barrier and it is very useful data for the experimentalist. Making a comparison with physical-organic chemistry, the height of the barrier can be estimated using the Hammond postulate, where the TS [in an exothermic reaction] is either close (very little activation of the molecules, TS is more like the reactant[s] than product[s]) or far (highly activated molecules, TS more like product[s] than reactant[s]). In a way our analysis is done a similar way as by another expert in the field, Prof. Odile Eisenstein. So, we located the H₂ activation TS, meaning that it of course corresponds to the H-H bond breaking with transfer of one H to the nitrogen. We followed the reaction coordinate (imaginary frequency) to get to the adduct (if any) and product of the reaction (the one reported in the profile). Then we analysed the geometry of the TS in order to understand the barrier.

c.) In Figure 10, structures are missing from step 2C to 2D by adding H₂ and B(Mes)₃. If the energetics of these steps are higher than the energy of 2D, then the current analysis and conclusions do not stand.

RESPONSE: This comment seems to miss the addition of $\text{H}_2/\text{B}(\text{Mes})_3$ in path **2D₂** and the $[\text{K}(\text{B15C5})_2]^+$ cation present throughout but not illustrated for brevity (its presence is implicit in the negative charge of **1** but we now clarify that in the figure 9 and 10 captions – note **2C** is neutral with the $\text{HB}(\text{Mes})_3$ in principle an ion pair with the K cation so if the $\text{HB}(\text{Mes})_3$ re-enters the primary coordination sphere of uranium in **2D₁** then the negative charge is reintroduced; conversely, neutral **2C** requires $\text{H}_2/\text{B}(\text{Mes})_3$ as shown to form **2D₂**). As mentioned in the text and in the figure 10, **2D_{1/2}** are transition states, and following the intrinsic reaction coordinates, in both cases, leads to a separation of the three molecules. There are no ‘adducts’ because we have FLP-type reactions, which makes sense.

d.) There are other reaction pathways are not explored, one intuitive one is the direct addition of H_2 to terminal nitrogen, etc.

RESPONSE: Outer-sphere addition of H_2 to even a highly basic nitride has never been observed in f-element chemistry. We have tried this calculation and every attempt yielded the TS reported in the manuscript, in line with a σ -bond metathesis reaction of H_2 .

2. Electronic structure analysis along reaction pathway is lacking. Hence, the conclusions about the f1-electron participation and oxidation state change along reaction pathways are not supported by the current reported data.

a.) The authors claim that the “G and H values are similar for the non-assisted mechanism” which is confusing to this reviewer. Once the number of speciation changes from reactants to products, the G and H values should change significantly, for example the steps from **1**+ H_2 to **A** and from **6** to **4**. It would be helpful to show the details how the G values are calculated in the supporting information.

RESPONSE: Using the harmonic approximation and frequency calculations, one can get the H and G values of all compounds. These values were reported in all profiles so our conclusions are based on this. For **6** to **4**, there is no entropy effect as the number of molecules on each side of the equation is the same, since **4** is a non-bonded ion pair. For **1**+ H_2 to **A**, it means that the main effect is a low thermodynamic stability of the H_2 adduct rather than an entropic effect. Also, recall that within the harmonic approximation the G values are questionable, explaining why our analysis is based on the H values. However, [REDACTED] the G values were provided and are in line with the H values.

b.) The authors reported NPA and spin density for direct addition mechanism and showed that there is no 5f1 electron participation along the reaction pathway. But similar analysis was not reported for the FLP mechanism. Therefore, the strong conclusion that 5f1 electron participation has no evidence to support.

RESPONSE: This is a fair point. [REDACTED]
We of course did the same analysis for the FLP reaction and found similar results. To compare directly with the direct addition to nitrogen, we analysed the density for the first H_2 activation in an FLP reactivity. In the three cases, there is spin depletion on the N (-0.12 on **1**, -0.13 on the adduct and -0.13 at the TS) whereas the unpaired spin density is clearly at the uranium centre (1.19 in **1**, 1.12 in the adduct and 1.13 at the TS). We have added these data into the discussion of the computed FLP-mechanism accordingly.

c.) Similarly, the authors conclude that the uranium center changes oxidation state along the reaction pathway and this reviewer finds no relevant data that can be interpreted as oxidation state change. It would be helpful if the authors report relevant the electronic structure data along the reaction pathway.

RESPONSE: Perhaps the reviewer missed this but in the text we clearly described the oxidation states at key points or we are referring to molecules where the oxidation state has been defined earlier on but we have added some reminders at this point.

As a minor point, the figure captions in the reaction mechanisms should be more explicit, for example, indicating what is the difference between the numbers in parenthesis in Figure 9 and Figure 10 and the bold ones with no parenthesis?

RESPONSE: We thought the figure axes were clear but have now added definitions in the figure captions.

Finally, was there any attempt at capturing any of the intermediates in the laboratory, or at following the reaction with any kind of spectroscopy?

RESPONSE: We looked at ReactIR and NMR spectroscopies but we encountered solubility issues since as we mentioned previously **6** is like brick-dust in all solvents (see previous responses). Also, **6** converts to **4** even without deliberate oxidation (see previous responses) so then tracking conversions reliably is sadly just not realistic because it is not possible to know what proportionately is in solution and what is not and therefore how representative any spectra (IR or NMR) actually are.

Reviewer #2:

However, before the manuscript to be published, this reviewer still has the following concerns and advise the authors to carefully consider. The choice of title phrasing and the efforts to establish synthetic cycle by irrelevant extra experiments all showed that the authors attempt to relate their work to ammonia synthesis by hydrogenation, despite the authors repeatedly argued in their response letter that this is not one of their priority claims. Therefore, this reviewer suggests that the authors should reconsider their choice of title and modify the wordings about "synthetic cycle" since the emphasis of ammonia synthesis adds little value to the manuscript but causes a lot of confusion and could be misleading to the reviewers and more importantly to the readers.

Major comments:

1) This reviewer doesn't see the point of putting "ammonia synthesis" in the beginning of the title. This should be removed as suggested by reviewer 3 from last round peer review. In their response letter, the authors also admitted that ammonia synthesis is never the priority point they claimed for this manuscript but the H₂ addition to terminal uranium nitride and FLP chemistry for actinides are. "Hydrogenations of terminal uranium(V)-nitride involving direct-addition or Frustrated Lewis Pair mechanisms" or so is more appropriate.

RESPONSE: We have changed the title to "*Terminal Uranium(V)-Nitride Hydrogenations Involving Direct Addition or Frustrated Lewis Pair Mechanisms*".

2) Relevantly, the last sentence in the abstract "A synthetic hydrogenation cycle for ammonia synthesis is demonstrated, and this work establishes a unique marriage of actinide and FLP chemistries." should be removed or replaced. Hydrogenation by definition means "a chemical reaction between molecular hydrogen (H₂) and another compound or element". In this report, the hydrogenation of terminal uranium(V) nitride only led to uranium amide but not uranium hydride and free ammonia. A genuine synthetic hydrogenation cycle for ammonia synthesis should start from N₂ and H₂ and end with NH₃. Here, N was from NaN₃; uranium nitride was generated from salt metathesis between U-Cl and NaN₃ in the presence of KC₈; the release of ammonia required treatment of U-NH₂ with Lewis acid Me₃SiCl followed by hydrolysis of Me₃SiNH₂. By all means, this should not be called "a synthetic hydrogenation cycle for ammonia synthesis" since only one step (M-nitride to M-NH₂) was achieved by hydrogenation and the rest involves multiple transformations and N source was NaN₃ but not N₂.

RESPONSE: This is a fair point. For the particular sentence that has been highlighted we have changed 'hydrogenation' to 'reactivity' and gone through the manuscript to tidy up anything related to either be a 'synthesis', 'reaction', 'reactivity', or 'overall hydrogenation' where we make it very clear what the individual steps are.

3) The authors claimed that "However, in contrast 1 reacts with H₂ under mild conditions despite the fact it is a high oxidation state metal and not of a low coordination number nor electron-rich as a 5f¹ metal ion." This is not true. First, U(V) should not be considered high oxidation state metal compared to Ru(IV) and Os(IV). Actually, Ru(IV) and Os(IV) is more oxidizing than U(V) in normal conditions. Secondly, taking account the size of metal ions, U(V) is much larger than Ru(IV) and Os(IV), therefore, the coordination number of 5 in uranium(V) complexes should not be considered "not low coordination number" while the coordination number of 4 is called low coordination number by the authors for Ru(IV) and Os(IV) nitride complexes. Indeed, in the Ru and Os cases, the metal nitride complexes

were able to evolve NH₃ but here in the U case, the authors were unable to achieve the hydrogenolysis of U-NH₂. About electron-rich or poor, 5f 1 metal ion is not necessarily less electron-rich than a d⁴ metal ion. For instance, Cu(III) is d⁸ metal ion but no one would call it electron-rich; while Th(III) is a 6d¹ metal ion but no one would call it not electron-rich. This all depends on the redox potentials as well as the orbital matching. Here, U(V) is more readily oxidizable and less likely to be reduced than Ru(IV) and Os(IV). Therefore, this reviewer doesn't see any point to call the U(V) in this study "a high oxidation state metal, not low coordination number, not electron-rich" compared to the literature precedents Ru(IV) and Os(IV).

RESPONSE: We have carefully considered what the reviewer has stated because it raises some interesting and very fundamental questions. However, whilst we see what they are getting at we cannot wholly agree with them. If a metal has oxidation states 0-6 available to it then 5 and 6 are at the high end. Compound **1** is therefore a high oxidation state uranium complex and that is separate and not to be confused with how oxidising it is since the reviewer is correct that that is a function of ligands and orbital energies so how oxidising they are is not relevant to labeling something as high, medium, or low oxidation state. If a metal can [formally] have 0-6 valence electrons and it only has 1 then it is by definition at the electron poor end just as if it can have 0-10 and it has 8 it is certainly electron rich. Compound **1** is therefore electron poor. When considering coordination numbers, 6 is quite general, most people consider 4 and below as low coordinate, especially for high oxidation state metals (for a recent consideration of this matter see Taylor and Kays, *Dalton Transactions* **2019**, doi:10.1039/c9dt02402f). Indeed, in reference 1 in the section on the reactivity of terminal metal-nitrides with H₂ square planar complexes are referred to as "low-coordinate", and that includes the Ru, Os, and Ir nitrides under discussion. Since complex **1** is five coordinate, it is not low-coordinate. We do not refer to what it is, just what it is not. Two things are also being conflated in the point above. If a metal is ML₅ (L = monodentate ligand) it is five coordinate, and therefore not low-coordinate, irrespective of whether M is lithium or uranium; the number of sites taken up is a separate concept to how proportionately large and saturated the metal is which is related to sterics. Given our above points, and since we feel this is really a matter of style/opinion, we feel we are allowed to have our opinion and let the reader decide for themselves. However, not being insensitive to the reviewer we have softened some of our phrases, for example in the one quoted we have inserted that the uranium "can be considered to be..." so it is not described as an absolute.

4) This reviewer disagrees with the authors about the reasons that not providing the full NMR spectra for compounds 3,4,5,8. We are in the digital era. In order to improve the reproducibility as well as the transparency of the published research, all possible characterization should be provided upon publication. This reviewer believes that the editors and the authors can work out a way to find a solution to include full ¹H NMR spectra for compounds 3,4,5,8. And if not, a specific reason for not able to obtaining a satisfactory clean NMR spectrum should be provided.

RESPONSE: We would like to emphasise that we did not disagree with the principle of the reviewer's request, but our prior point was based on prior experience at this journal, where previously because it was not practicable to call out every SI figure individually we [frustratingly for us] had to remove NMR spectra from the SI of another paper at this journal to meet the rigid editorial house-style. However, we are delighted to say that the handling editor has now informed us that *Nature Communications* has recently relaxed its rules on this matter, since it was recognised by Springer that chemistry papers in particular were suffering from this inflexibility, of which our paper was just one such example. The solution we have come up with is that the figures are now in the SI and they are called out in the data availability section to avoid cumbersome prose calling them all out. We thus include figures for **3**, **4**, and **8** but unfortunately cannot include a spectrum for **5** because of solubility issues and this coupled to its radical nature (note the EPR data are clear – the unpaired spin density is delocalised right across the BMes₃ moiety) means the provenance of any spectrum recorded is uncertain and so would be unsound to include.

---End---

Reviewers' comments:

Reviewer #1 (Remarks to the Author):

The response did not answer the questions raised by this reviewer; hence, this manuscript is not suggested for publication in its current form.

1. For any specific scientific questions, showing the actual data is the only objective way that can lead to objective conclusions. The statement about 'go-to' expert is an opinion not a verifiable fact.
2. Regarding question 1.a), the response directly contradicts with the similar 2B structure in Figure 10.
3. Regarding question 1.c), the authors missed the point. The structures of the adducts of H₂ and BMe₃ are not reported. 2D1 and 2D2 are transition states not adducts. In Figure 9, the formation of adduct A (by adding H₂ only) needs 8.3 Kcal/mol. How much more energy is needed to form the adducts of H₂ and BMe₃ in Figure 10?
4. Regarding questions 2.b) and 2.c), the newly added spin-density data (1.19 in 1, 1.12 in the adduct and 1.13 at the TS) by the authors proves that the oxidation state of metal actually does not change along reaction pathway. Similarly, this set of data also proves that the 5f electron does not participate the activation of H₂ since the spin-density of the metal center stays roughly the same in transition state compared to the reactants and products.

Reviewer #2 (Remarks to the Author):

I am fine with the current version of manuscript. The authors have addressed most of my concerns or respond with a fair argument.

Reviewer #3 (Remarks to the Author):

Review

The manuscript details the reaction of dihydrogen with a terminal U(V) nitride occurs and in the presence of boranes, such as BPh₃ and B(Mes)₃. The experimental data are well analysed and conclusive. However the computational part is fuzzy, despite the well-founded requests for clarifications by the previous referees. Being in the field of actinide computational chemistry is not an excuse for being quick and careless in the analysis and presentation of computational results! The authors seem to have deliberately walked around clarifying a number of points as detailed below.

As it is reported and read in the current manuscript, I am not at all satisfied with the level of details to reach publication standard, in particular as the computational chemistry part should serve the goal of proposing chemical mechanisms, and kinetics/energetics confirmation. The reaction of uranium (V) nitride species implies oxidation state changes which, in the absence of data on spin density, atomic charges, bond analysis for ALL intermediate and TS not just for complexes 2A, 2B of Figure 9. There is plenty of space in the supplementary material, to prepare a table with spin, atomic charges, U-N lengths, bond orders, spin multiplicities and spin contamination. To be more specific:

1. I would like to have quantitative elements on the oxidation state along figure 9 and 10 reaction paths, through uranium spin density and charge for all complexes. The computational part of this study is needed to provide sound evidence for the oxidation state change from (V).

2. To get a feeling of the curvature at the TS saddle point, it is common to report the identified imaginary frequency. This must be added to the supplemental table.

3. [REDACTED]

Furthermore, the beginning the the "Computational reaction mechanism profiles", page 12, only mentions DFT (B3PW91). What happened to the solvent and dispersion corrections ???

4. Another unclear technical aspect, concerns the multiplicity enforced for each complexes ? Are these unrestricted calculations on a doublet potential energy surface ? The spin multiplicity of each stationary point must be included, along with spin contamination, the latter giving indications about potential multireference effects.

5. I understand that it is out of reach to include spin-orbit interaction, but the authors may estimate its contribution based on the known spin-orbit effects for atomic uranium in the various oxidation states.

6. It is repeated a few times that the reason for the reactivity of U(V) nitride has to do with the unpaired uranium 5f being not purely non-binding? What about looking at the atomic character of the spin-localized orbital in the reactant. Such information can be found in the NPA analysis and also by drawing the orbital.

7. Experimentally, no evidence for the uranium(VI)-nitride and H₂. What about trying to look for the homologous reaction path of figure 10 for the uranyl(VI). This seems necessary to help rationalize the reason behind the uranyl(V) reactivity.

From my comments related to the computational part of this manuscript, I request major changes and clarification. Despite the novelty of this study regarding uranium chemistry, the computational part needs work to reach the publication standards of Nature communications.

Reviewer Comments and Responses:

Reviewer #1:

The response did not answer the questions raised by this reviewer; hence, this manuscript is not suggested for publication in its current form.

1. For any specific scientific questions, showing the actual data is the only objective way that can lead to objective conclusions. The statement about 'go-to' expert is an opinion not a verifiable fact.

RESPONSE: We have provided all data when requested.

2. Regarding question 1.a), the response directly contradicts with the similar 2B structure in Figure 10.

RESPONSE: As we pointed out in our previous response, H₂ cannot bind end-on to U, what it can do is bind side-on to uranium or end-on to the nitride. However, it only binds end-on to the nitride when the borane is present. If the borane is not present then the side-on form is obtained, always, so yes at that point A is lowest. It is also worth pointing out that A is not a TS, it is a shoulder on the way to the TS B, but 2B is a TS so they are not equivalent as implied in the line of questioning. There are no other species on the PES, so it is what it is.

3. Regarding question 1.c), the authors missed the point. The structures of the adducts of H₂ and BMe₃ are not reported. 2D1 and 2D2 are transition states not adducts. In Figure 9, the formation of adduct A (by adding H₂ only) needs 8.3 Kcal/mol. How much more energy is needed to form the adducts of H₂ and BMe₃ in Figure 10?

RESPONSE: With respect, we have not missed the point. As we stated before, *there are no adducts because of the FLP nature of the chemistry*; we performed exhaustive searches but no stable adducts were located on the PES, neither for the U(V) system **1** or the U(VI) system **2** which we have now added to the SI in response to Reviewer #3 (see below). In every case the three molecules just go away from each other. So there are no adducts to report, not because we didn't check but because the calculations suggest that they don't exist. Therefore, it is not possible to answer the adduct energy question because that is asking about the energy of something that doesn't exist.

4. Regarding questions 2.b) and 2.c), the newly added spin-density data (1.19 in **1**, 1.12 in the adduct and 1.13 at the TS) by the authors proves that the oxidation state of metal actually does not change along reaction pathway. Similarly, this set of data also proves that the 5f electron does not participate the activation of H₂ since the spin-density of the metal center stays roughly the same in transition state compared to the reactants and products.

RESPONSE: The reviewer is correct and what was written in the manuscript. We have re-read this carefully a few times now to check and have concluded that the text is an accurate and clear representation of the reviewer's summary.

Reviewer #2:

I am fine with the current version of manuscript. The authors have addressed most of my concerns or respond with a fair argument.

RESPONSE: We thank the reviewer for their considered comments, which have improved the manuscript, and also for keeping an open mind and considering carefully in return our arguments.

Reviewer #3:

The manuscript details the reaction of dihydrogen with a terminal U(V) nitride occurs and in the presence of boranes, such as BPh₃ and B(Mes)₃. The experimental data are well analysed and conclusive. However the computational part is fuzzy, despite the well-founded requests for clarifications by the previous referees. Being in the field of actinide computational chemistry is not an excuse for being quick and careless in the analysis and presentation of computational results! The authors seem to have deliberately walked around clarifying a number of points as detailed below. As it is reported and read in the current manuscript, I am not at all satisfied with the level of details to reach publication standard, in particular as the computational chemistry part should serve the goal of proposing chemical mechanisms, and kinetics/energetics confirmation. The reaction of uranium (V) nitride species implies oxidation state changes which, in the absence of data on spin density, atomic charges, bond analysis for ALL intermediate and TS not just for complexes 2A, 2B of Figure 9. There is plenty of space in the supplementary material, to prepare a table with spin, atomic charges, U-N lengths, bond orders, spin multiplicities and spin contamination. To be more specific:

RESPONSE: We have added the requested data in tables in the SI.

1. I would like to have quantitative elements on the oxidation state along figure 9 and 10 reaction paths, through uranium spin density and charge for all complexes. The computational part of this study is needed to provide sound evidence for the oxidation state change from (V).

RESPONSE: This information is now compiled in the SI and reflected in the discussion in the manuscript where the change in uranium oxidation state (last step) is clear and in-line with the computed data in the tables.

2. To get a feeling of the curvature at the TS saddle point, it is common to report the identified imaginary frequency. This must be added to the supplemental table.

RESPONSE: This data has been added into the SI tables, and they are around 1000 cm^{-1} , which is in-line with classical H_2 activation.

3. [REDACTED]
[REDACTED]
[REDACTED] Furthermore, the beginning the the "Computational reaction mechanism profiles", page 12, only mentions DFT (B3PW91). What happened to the solvent and dispersion corrections ???

RESPONSE: Fair point, this was simply missing text, which we apologise for, so we have included appropriate text in the computational description of the SI and at the start of the computational section of the manuscript.

4. Another unclear technical aspect, concerns the multiplicity enforced for each complexes ? Are these unrestricted calculations on a doublet potential energy surface ? The spin multiplicity of each stationary point must be included, along with spin contamination, the latter giving indications about potential multireference effects.

RESPONSE: Indeed these are doublet PESs. This information is now in the additional SI tables. The spin contaminations are also given and they are quite small indicating that our results are correct. Spin multiplicities are also included in the new SI tables.

5. I understand that it is out of reach to include spin-orbit interaction, but the authors may estimate its contribution base on the known so effects for atomic uranium in the various oxidation states.

RESPONSE: Indeed, it is out of reach. However, from reference 38 we know it is 1860 cm^{-1} for U(V) in our nitrides and that work is referenced clearly if the reader wants to delve into the wider matter further.

6. It is repeated a few times that the reason for the reactivity of U(V) nitride has to do with the unpaired uranium 5f being not purely non-binding? What about looking at the atomic character of the spin-localized orbital in the reactant. Such information can be found in the NPA analysis and also by drawing the orbital.

RESPONSE: A thorough dissection of our uranium-nitrides was published in reference 38. In a simplified case it is a pure 5f phi orbital, but in reality mixing produces a m_l of $m_l 2$ and 3 composite character; indeed it is the CF of the nitride that introduces mixing and leads to a highly nuanced electronic structure. That in itself shows the 5f electron is not a simple non-bonding electron, and this point is already made and referenced on page 15 of the manuscript.

7. Experimentally, no evidence the uranium(VI)-nitride and H₂. What about trying to look for the homologous reaction path of figure 10 for the uranyl(vi). This seems necessary to help rationalize the reason behind the uranyl(V) reactivity.

RESPONSE: The calculations have been done and the resulting profile is now in the SI. It confirms what we found experimentally, which is that **2** does not react with H₂ even in an FLP scenario, with extremely high barriers to overcome. A brief mention of this is now made at the beginning of the computational section of the manuscript referring the reader to the SI. We have also added a table of computed data in the SI for this hypothetical U(VI) profile to match the U(V) ones corresponding to figs 9 and 10.

From my comments related to the computational part of this manuscript, I request major changes and clarification. Despite the novelty of this study regarding uranium chemistry, the computational part needs work to reach the publication standards of Nature communications.

RESPONSE: We hope the reviewer can conclude that we have done everything that they have requested, as indeed we have tried to do at every stage of review.

---End---

REVIEWERS' COMMENTS:

Reviewer #3 (Remarks to the Author):

The authors have properly addressed the concerns and responded with fair arguments. I appreciate the additional data placed in the Supporting Information. However, for the sake of rigour, Tables 23-25 should be properly labelled and cleaned up:

- Place only 2-3 digits for the values of spin density, charge, Wiberg bond index, frequency.
- Add units to the UN bond length (Å) and (cm⁻¹) to the imaginary frequency, which should be relabelled "TS imaginary frequency".

With these modifications, I recommend publication.

Reviewer Comments:

The authors have properly addressed the concerns and responded with fair arguments. I appreciate the additional data placed in the Supporting Information. However, for the sake of rigour, Tables 23-25 should be properly labelled and cleaned up:

- Place only 2-3 digits for the values of spin density, charge, Wiberg bond index, frequency.

RESPONSE: We have trimmed data to 3dp. We feel this is reasonable given the nature of the work. We have left the bond lengths to further dp consistent with that often determined by SC-XRD.

- Add units to the UN bond length (Å) and (cm⁻¹) to the imaginary frequency, which should be relabelled "TS imaginary frequency".

RESPONSE: These additions and corrections have been made.

With these modifications, I recommend publication.

RESPONSE: Many thanks.